# Exciton engineering of 2D Ruddlesden–Popper perovskites by synergistically tuning the intra and interlayer structures

Songhao Guo[1], Willa Mihalyi-Koch [ID][2], Yuhong Mao[1], Xinyu Li[3], Kejun Bu [ID][1], Huilong Hong [ID][3], Matthew P. Hautzinger [ID][2], Hui Luo[1], Dong Wang [ID][1], Jiazhen Gu[3], Yifan Zhang[1], Dongzhou Zhang [ID][4], Qingyang Hu [ID][1], Yang Ding [ID][1], Wenge Yang [ID][1], Yongping Fu [ID][3] ✉, Song Jin [ID][2] ✉ & Xujie Lü [ID][1] ✉

Designing two-dimensional halide perovskites for high-performance optoelectronic applications requires deep understanding of the structure-property relationship that governs their excitonic behaviors. However, a design framework that considers both intra and interlayer structures modified by the A-site and spacer cations, respectively, has not been developed. Here, we use pressure to synergistically tune the intra and interlayer structures and uncover the structural modulations that result in improved optoelectronic performance. Under applied pressure, $(BA)_2(GA)Pb_2I_7$ exhibits a 72-fold boost of photoluminescence and 10-fold increase of photoconductivity. Based on the observed structural change, we introduce a structural descriptor $\chi$ that describes both the intra and interlayer characteristics and establish a general quantitative relationship between $\chi$ and photoluminescence quantum yield: smaller $\chi$ correlates with minimized trapped excitons and more efficient emission from free excitons. Building on this principle, we design a perovskite $(CMA)_2(FA)Pb_2I_7$ that exhibits a small $\chi$ and an impressive photoluminescence quantum yield of 59.3%.

Two-dimensional (2D) Ruddlesden-Popper (RP) lead halide perovskites have emerged as promising multi-functional 2D semiconductors, characterized by the formula $(LA)_2(A)_{n-1}Pb_nX_{3n+1}$. Here, LA represents a long-chain organic interlayer (spacer) cation, while A denotes a small intralayer cation (in the perovskite cage), X is a halide anion, and $n$ is an integer indicating the number of perovskite layer[1,2]. The multiple-quantum-well structure in 2D RP perovskites introduces significant quantum and dielectric confinements. Consequently, the main excited-state carrier species are strongly bound excitons

(electron-hole pairs) with binding energies of hundreds of milli-electron volts (meV), which dictate the optical and optoelectronic properties[3,4]. Such unique excitonic feature makes them a diverse and versatile material system for exploring excitonic physics and technological applications[5,6].

Furthermore, 2D RP perovskites are promising for photovoltaics and optoelectronics due to improved stability and higher structural and compositional tunability compared to 3D lead halide perovskites[7–10]. Thus far, solar cells based on 2D perovskites reached an

[1]Center for High Pressure Science and Technology Advanced Research (HPSTAR), Shanghai, China. [2]Department of Chemistry, University of Wisconsin-Madison, Madison, WI, USA. [3]Beijing National Laboratory for Molecular Science, College of Chemistry and Molecular Engineering, Peking University, Beijing, China. [4]Hawaii Institute of Geophysics & Planetology, University of Hawaii Manoa, Honolulu, HI, USA. ✉e-mail: yfu@pku.edu.cn; jin@chem.wisc.edu; xujie.lu@hpstar.ac.cn

energy conversion efficiency of 18.3%[11], lower than the efficiency of 25.7% for 3D halide perovskites[12]. Light-emitting diodes based on 2D perovskites show an external quantum efficiency up to 20.5%, leaving large room for further improvement[13,14]. These unsatisfactory efficiencies are often attributed to overly localized and bound excitons, which promote nonradiative recombination and suppress carrier transport[15,16]. Therefore, to enable high-performance solar cells, photodetectors, and light-emitting diodes, it is necessary to understand the structural features that govern the excitonic behavior in 2D perovskites and develop materials design principles for engineering the desired excitonic features.

Recently, extensive efforts have been made to tailor the interlayer LA cations in 2D perovskites with $n = 1$[16–18] and correlate their optical properties to several structural descriptors, including those describing intra-octahedral distortions and octahedral tilting[19–21]. However, the $n > 1$ RP perovskites are more structurally complex because they possess an additional degree of freedom through the intralayer A cations that can also modulate the excitonic properties[22–24]. Existing parameters that are used to describe the structural consequence of incorporated A cations in 3D perovskites, such as the Goldschmidt tolerance factor[25], do not consider the interaction of both the LA and A cations with the inorganic framework. The collective effect of the intra and interlayer structural features on excitonic behavior of these higher $n$ RP perovskites remains unclear. To achieve the most desirable excitonic properties in the $n > 1$ perovskites, the intra and interlayer structures need to be considered together and synergistically tuned. Achieving a deeper understanding requires an efficient tuning knob that allows for continuous regulation of the anisotropic structures in combination with advanced in situ diagnostic tools.

As a state parameter, pressure has been widely used to modify the structural and electronic properties of materials without compositional modification[26–28]. High-pressure research enables us to reach structural regimes that are otherwise inaccessible and elucidate structure-property relationships[29–31]. However, previous high pressure studies of 2D perovskites mostly focused on $n = 1$ perovskites and they have not studied different A cations in perovskite cages[32–35], therefore could not have investigated the synergistic effects of the intra and interlayer structural characteristics. Here, by applying pressure to 2D $n = 2$ RP perovskites, we synergistically tune both the intra and interlayer structures to engineer the excitonic properties and observe concurrent enhancement of photoluminescence (PL) and photoconductivity properties. Pressure controllably modulates the carrier dynamics in $(BA)_2(GA)Pb_2I_7$ (BA = butylammonium and GA = guanidinium) from competition between trapped and free excitons to competition between free excitons and free carriers, such that its PL intensity is boosted by 72-fold upon compression to 2.1 GPa and its photoconductivity is also enhanced by 10-fold arising from the dissociated excitons. Based on these structural changes, we introduce a structural descriptor $\chi$ that considers the effects of both the intra and interlayer structural characteristics in 2D RP ($n > 1$) perovskites and can be generally correlated to the excitonic properties across many classes of 2D perovskites with different LA and A cations and $n$ numbers. Guided by the design principles uncovered from our high-pressure study, we design and synthesize a 2D perovskite with highly favorable excitonic properties, achieving an impressive PL quantum yield (PLQY) among known $n = 2$ RP lead iodide perovskite materials.

## Results
### Pressure-modulated excitonic properties
Owing to the soft and polar nature of the Pb-X frameworks, excitons in 2D perovskites strongly couple to phonons and are often locally trapped (Fig. 1a)[4,17]. The asymmetric PL peak commonly observed in 2D RP perovskites stems from the dynamic equilibrium between free excitons and trapped excitons, where the former contribute to the primary PL peak and the latter manifest as a low-energy tail[36,37]. Exciton

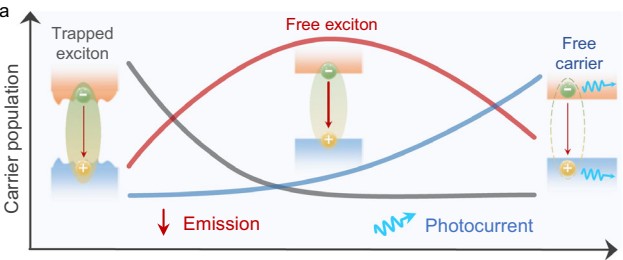

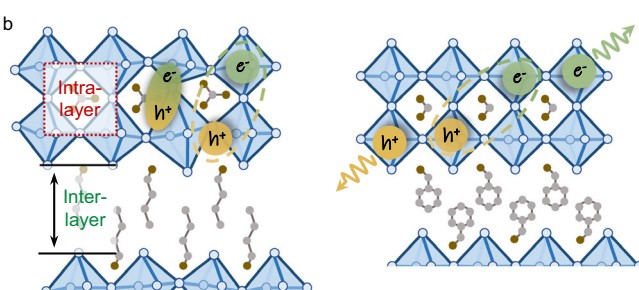

**Fig. 1 | Exciton behaviors and intra/interlayer structural characteristics of 2D RP perovskites. a** Schematic diagram of carrier dynamics in 2D RP perovskite with various excitonic features. The trapped excitons in 2D RP perovskites with distorted [PbI$_6$] octahedra exhibit weak emission, which would transform to free excitons with strong emission by suppressing exciton trapping. With the reduction of exciton binding energy, the free excitons could further dissociate into free carriers, leading to high photocurrent. **b** Illustrations of the synergistic manipulation of the intra and interlayer structures towards more desirable excitonic properties of 2D RP perovskites (using the example of $n = 2$).

trapping also induces more phonon-assisted nonradiative recombination pathways that lead to PL quenching[38]. Furthermore, 2D perovskites tend to exhibit less efficient separation and migration of excited carriers (Fig. 1a) than 3D perovskites, which is required for photovoltaic and photodetection (PD) applications[39]. Both the strong exciton-phonon coupling and large exciton binding energy are unfavorable for the optoelectronic performances of 2D perovskites. To achieve high PL and PD performances of the more complex $n > 1$ RP perovskites, the effects of both the intra and interlayer structures must be considered together, necessitating the development of an updated structural parameter. As shown in Fig. 1b, the layers composed of [PbI$_6$]$^{4-}$ octahedra and A cations are termed as intralayer, whereas the LA cations situated between two neighboring [PbI$_6$]$^{4-}$ octahedral layers are referred as interlayer. Manipulating the structure in a controlled manner, e.g., using pressure processing, can help elucidate the synergistic effects of intra and interlayer cations on the excitonic properties of $n > 1$ RP perovskites.

To uncover the intra and interlayer structural features that govern the excitonic properties, we start by examining the effect of pressure modulation on photoluminescence and photoconductivity of the 2D RP perovskite $(BA)_2(GA)Pb_2I_7$. We choose this perovskite because it incorporates an oversized A-cation (GA) into the perovskite cage, resulting in considerable octahedral distortion and a high level of exciton trapping and asymmetric PL peak. Additionally, the LA cations (BA) are flexible alkyl chains that make the interlayer highly compressible. Therefore, $(BA)_2(GA)Pb_2I_7$ is an ideal perovskite to investigate whether pressure-induced structural changes can improve excitonic behavior and analyze the nuanced structural-property relationships across a vast parameter space.

We first collected the in situ steady-state PL spectra of $(BA)_2(GA)Pb_2I_7$ under pressure regulation to examine the evolution of the excitonic emission peak (Fig. 2a). The PL intensity sharply increases during compression and reaches a 72-fold enhancement at 2.1 GPa compared

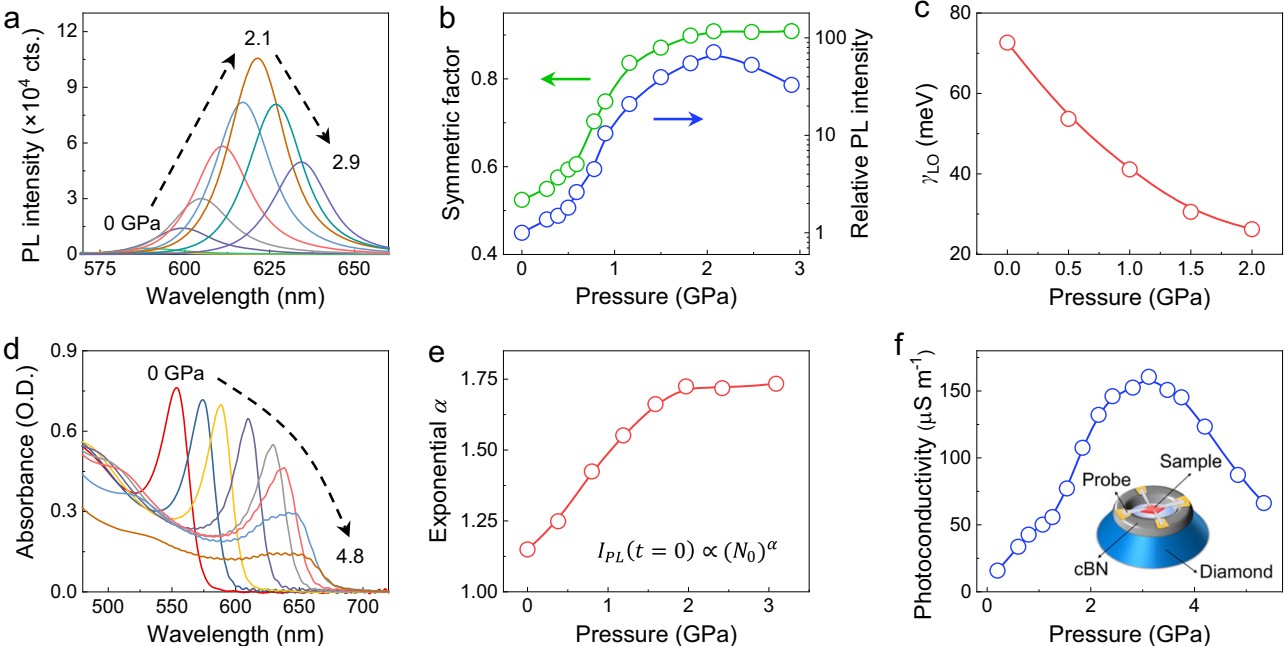

**Fig. 2 | Manipulating the exciton behavior and optoelectronic properties in (BA)$_2$(GA)Pb$_2$I$_7$ by pressure-tuning. a** In situ steady-state PL spectra at selected pressures, where a remarkably enhanced emission with a more symmetric PL line shape is observed during compression, reaching the maximum at 2.1 GPa. **b** Symmetric factor and the relative PL intensity (in the logarithmic scale) as a function of pressure. The PL intensity reaches the maximum when the symmetric factor approaches 1. **c** The fitted exciton-phonon coupling strength ($\gamma_{LO}$) as a function of pressure. **d** Absorption spectra of an exfoliated (BA)$_2$(GA)Pb$_2$I$_7$ single crystal at different pressures, where the gradually weakened excitonic absorption

peak indicates the reduction of exciton binding energy $E_b$. **e** The exponential value $\alpha$ of carrier density dependent initial PL intensity ($I_{PL}$ at delay time $t = 0$ ps) as a function of pressure. The plot follows the law of $I_{PL}(t = 0) \propto (N_0)^{\alpha}$, and the $\alpha$ value indicates the major carrier specie ($\alpha = 1$ implies excitons, $\alpha = 2$ implies free carriers, and $\alpha$ between 1 and 2 suggests the coexisting of excitons and free carriers). **f** The photoconductivity at different pressures, where a significant 10-fold enhancement is achieved at 3.1 GPa. The inset shows a schematic illustration of the photocurrent measurement setup within a diamond anvil cell. Source data are provided as a Source Data file.

to the initial value. Note that the tail of the PL peak at low-energy side, which is ascribed to the radiative recombination of trapped excitons, gradually disappears, leading to a symmetric PL line-shape resulting from free excitons at 2.1 GPa. It is well recognized that the emission performance of 2D perovskites is closely correlated to the shape of the PL peak[38,40]. Because it is more difficult to fairly compare the absolute PL intensity of different perovskite samples under various conditions, we introduce a symmetric factor $S$ to quantify the PL tailing effect:

$$S = \frac{\int_0^{\lambda_{peak}} I(\lambda) d\lambda}{\int_{\lambda_{peak}}^{+\infty} I(\lambda) d\lambda} \quad (1)$$

where $I(\lambda)$ is the PL intensity at the given wavelength and $\lambda_{peak}$ is the PL peak position. As shown in Fig. 2b and Supplementary Fig. 1, the symmetric factor increases with increasing pressure and exhibits a linear correlation with PL intensity in the logarithmic scale. Furthermore, the fluorescence images captured at various pressures clearly depict the variations of emission brightness. (Supplementary Fig. 2).

To quantify the strength of exciton-phonon coupling in (BA)$_2$(GA)Pb$_2$I$_7$, temperature-dependent PL measurements were performed at selected pressures. PL peak broadening can be observed due to the enhanced phonon scattering as temperature increases[16,22]. Supplementary Fig. 3 shows the PL line widths at different pressures and temperatures for (BA)$_2$(GA)Pb$_2$I$_7$, from which we conclude that thermal broadening is suppressed upon compression. We fitted the PL line width as a function of temperature to determine the coupling strength $\gamma_{LO}$ at selected pressures (Supplementary Fig. 4, see detailed description in Supplementary Information). Fig. 2c shows that the exciton-phonon coupling strength $\gamma_{LO}$ decreases from 73 meV at

ambient conditions to 26 meV at 2 GPa. Consequently, the population of free excitons increases with increasing pressure, resulting in enhanced excitonic emission.

Besides the exciton-phonon coupling, the exciton binding energy $E_b$ strongly influences the excitonic properties and determines the possibilities of exciton recombination and dissociation[41,42]. To examine the evolution of $E_b$, we collected the in situ absorption spectra of an exfoliated (BA)$_2$(GA)Pb$_2$I$_7$ single crystal under high pressure. At ambient conditions, an excitonic absorption peak can be observed at 562 nm (Fig. 2d). Upon compression, the peak gradually redshifts, weakens, and disappears at around 5 GPa, indicating the reduction of $E_b$. The relative population of excitons and free carriers is described using:

$$I_{PL}(t = 0) \propto (N_0)^{\alpha} \quad (2)$$

where $I_{PL}(t = 0)$ is the PL intensity at time $t = 0$, $N_0$ is the carrier density, and $\alpha$ is the exponential value[43]. $\alpha = 1$ implies excitons, $\alpha = 2$ implies free carriers, and $\alpha$ between 1 and 2 suggests the coexisting of excitons and free carriers[44]. We conducted time-resolved PL spectra as a function of carrier density $N_0$ for (BA)$_2$(GA)Pb$_2$I$_7$ at different pressures (Supplementary Fig. 5), and fitted the $I_{PL}(t = 0)$ under different carrier density $N_0$ according to Equation 2. As shown in Supplementary Fig. 6, the initial exponential value $\alpha$ of 1.1 at ambient pressure suggests that PL in (BA)$_2$(GA)Pb$_2$I$_7$ mainly originates from excitonic emission, which can be further validated by the constant quantum efficiency under low carrier density (Supplementary Figs. 7 and 8). Upon compression, the $\alpha$ value increases to 1.7 at 3.1 GPa (Fig. 2e), suggesting the increase of free carriers due to the pressure-induced reduction of $E_b$ which influences the carrier dynamics involving the dissociation and recombination

processes. The oscillator strength of the exciton emission undergoes significant modulation since it is proportional to the binding energy[5,45]. Consequently, the radiative recombination of excitons becomes more efficient, enabling it to outperform non-radiative recombination, thereby leading to an increased PL intensity.

Moreover, the efficiency roll-off threshold of $(BA)_2(GA)Pb_2I_7$ is increased by more than one order of magnitude upon compression (Supplementary Fig. 9), indicating a suppressed Auger recombination due to the weakening of Coulombic interaction. This behavior permits high brightness under high carrier density which is preferred in the light emitting devices[8]. It is worth noting that the PLQY of $(BA)_2(GA)Pb_2I_7$ starts to increase as the injected carrier density rises at high pressure due to saturation of defects. This phenomenon occurs because excitons tend to be more readily captured by defects when they become delocalized under compression. This behavior indicates the transition of main carrier species from the equilibrium between trapped and free excitons to the equilibrium between free excitons and free carriers through the reduction of $E_b$[15,44]. We further performed in situ photoconductivity measurements on a $(BA)_2(GA)Pb_2I_7$ single-crystal flake under high pressures. The photocurrents exhibit fast on-off switch responses to light illumination and gradually increase with pressure up to ~3 GPa (Supplementary Fig. 10). This corresponds to a 10-fold increase of photoconductivity from 15.8 μS m⁻¹ at 0.2 GPa to 160.6 μS m⁻¹ at 3.1 GPa (Fig. 2f). These in situ measurements on $(BA)_2(GA)Pb_2I_7$ show that by applying pressure, we can transmute the carrier behavior from trapped excitons to free excitons and free carriers, which simultaneously improves the PL and photoconductivity performances of the 2D RP perovskite.

## Synergistically tuning intra and interlayer structures

To understand the pressure-induced structural and electronic changes that lead to the enhanced optoelectronic properties, we performed in situ single-crystal X-ray diffraction (Supplementary Fig. 11) and first-principles calculations. $(BA)_2(GA)Pb_2I_7$ adopts the $C2/c$ space group at 0.4 GPa with the lattice parameters of $a = 37.860$ Å, $b = 9.1394$ Å, $c = 8.8393$ Å, and $\beta = 93.41°$ (Supplementary Table 1). In $n = 2$ RP perovskites, the inorganic sublattice composed of two-layers of corner-shared $[PbI_6]^{4-}$ octahedra (aka, the perovskite cage) are separated by organic BA cations (Fig. 3a). The large GA cation sitting within the perovskite cage results in substantial distortion of $[PbI_6]^{4-}$ octahedra, leading to localized excitons[22,24]. The plots showing the variations of lattice constants with pressure (Fig. 3b) clearly reveal anisotropic compressibility. The contraction of interlayer $a$ lattice constant is much greater than that of the intralayer $b$ and $c$ lattice constants owing to the highly compressible nature of BA organic spacer cations: the interlayer distance ($L$) decreases from 12.5 Å at 0.4 GPa to 11.4 Å at 5.5 GPa (Fig. 3c). The lattice compression also shortens the Pb-I bond and shrinks the perovskite cage. Therefore, the volume of the perovskite cage ($V$) decreases from 268 Å³ at 0.4 GPa to 221 Å³ at 5.5 GPa (Fig. 3c). The interlayer distance and perovskite cage volume are schematically illustrated in Fig. 3a and their determination are described in detail in the Supplementary Information.

Theoretical calculations were performed to evaluate the evolution of electronic band structures based on modulated $(BA)_2(GA)Pb_2I_7$ crystal structures under high pressure (Fig. 3d and Supplementary Fig. 12). The more dispersive nature near the band edges under a higher pressure (2.9 GPa) suggests an enhanced carrier mobility. The exciton reduced mass decreases substantially from 0.17 at 0.4 GPa to 0.04 at 5.5 GPa (Supplementary Fig. 13). The $E_b$, which is calculated according to the Wannier-Mott exciton model[46] (see details in Supplementary Information), decreases from 187 meV at 0.4 GPa to 33 meV at 5.5 GPa (Fig. 3e). Impressively, such a reduced $E_b$ reaches the regime of $E_b$ for typical 3D halide perovskites and is comparable to the thermal energy at room temperature (26 meV).

Therefore, the pressure-induced structural modulation results in the suppression of trapped excitons and promotes the further

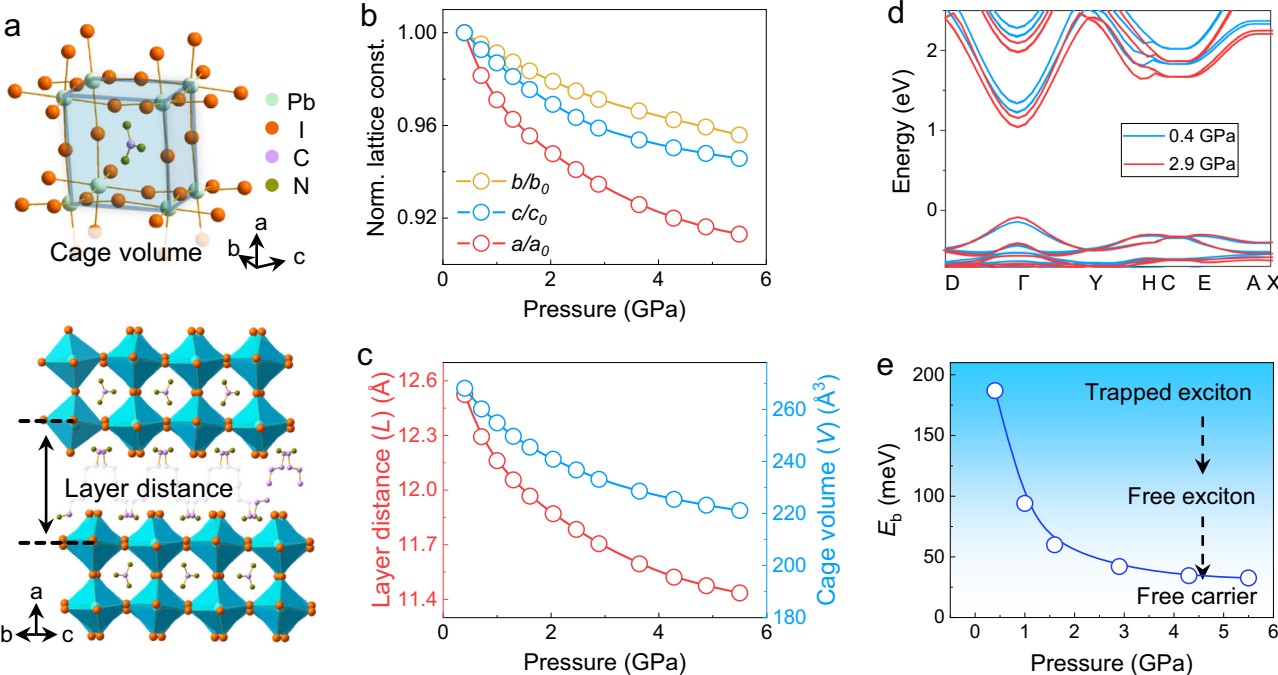

**Fig. 3 | Pressure-induced evolution of structural and electronic properties in $(BA)_2(GA)Pb_2I_7$. a** Crystal structure, which shows the occupation of the GA cation in the perovskite cage formed by eight $[PbI_6]^{4-}$ octahedra and the layered structure with interlayer BA cations. The perovskite cage volume and interlayer distance are also schematically illustrated. **b** The variations of lattice constants under high pressure along different crystallographic axes, revealing the anisotropic lattice compressibility. **c** Layer distance ($L$, left axis) and perovskite cage volume ($V$, right axis) as a function of pressure. **d** Calculated electronic structures at 0.4 and 2.9 GPa, which reveal a more dispersive nature near the band edges and suggest an enhanced carrier mobility at a higher pressure. **e** Calculated exciton binding energy $E_b$ based on structural evolution as a function of pressure. Source data are provided as a Source Data file.

dissociation of free excitons, which enhances PL and photo-conductivity performance. Such structural change is a combination of intra and interlayer effects. Within the intralayer, there is a decrease in the Pb-I bond length and an increase in the Pb-I-Pb bond angle under compression, resulting in a reduction in the perovskite cage volume and the idealization of the perovskite framework. Consequently, the bandgap narrows due to the increased overlap between Pb($6s$) and I($5p$) orbitals. In addition, these structural variations suppress exciton trapping through phonon hardening[24]. Furthermore, interlayer distance decreases upon compression which leads to increased packing density of spacer cations and the weakening of the dielectric confinement and consequently reduces the exciton binding energy[2,47].

## Proposing structural descriptor $\chi$ and materials design

To quantitatively and comprehensively describe the effects of structural modulation on the excitonic behavior of 2D RP perovskites, we introduce a structural descriptor $\chi$ that takes into account both intra and interlayer structural parameters:

$$\chi = V \times \frac{L}{\sqrt{N}} \qquad (3)$$

where $V$ and $L$ refer to the perovskite cage volume and interlayer distance, respectively, and $N$ is the number of non-hydrogen atoms in the interlayer spacer cation (more detailed discussion in the Supplementary Information). For example, the $\chi$ value for $(BA)_2(GA)Pb_2I_7$ at ambient condition is $0.158\,nm^4$, which decreases to $0.126\,nm^4$ at 2.0 GPa as a result of both intra and interlayer compression. Since the symmetric factor of PL peak ($S$) reflects the degree of exciton trapping and correlates well with the PL intensity (Supplementary Fig. 1), we plot the symmetric factor of $(BA)_2(GA)Pb_2I_7$ against the structural descriptor $\chi$ in Fig. 4a (blue squares). As $\chi$ decreases during compression, the

symmetric factor gradually increases, accompanied with the PL enhancement.

This structural descriptor $\chi$ unifies the intra and interlayer structural parameters using cage volume and layer distance in a balanced fashion that is normalized by the size of the interlayer spacer cation. The step-by-step correlation analysis for arriving at the structural descriptor $\chi$ is shown in Supplementary Figs. 14 and 15, along with the detailed description in Supplementary Information. Because the symmetry factor of PL peak does not depend on the specific details of the PL intensity, we can easily compare the data from the pressure regulation and chemical tailoring of many families of 2D RP perovskites. Interestingly, as shown in Fig. 4a, we found that $\chi$ empirically correlates well with the symmetric factor of the PL peaks for 40 cases of $n > 1$ 2D RP perovskites under pressure tuning or with different compositions of six different aliphatic alkylammonium LA cations and six different A cations (Fig. 4b)[23,24,38,40,48–50]. Thus, we believe $\chi$ can serve as an effective predictive parameter for structural design towards high-performance 2D RP perovskites. Importantly, each component of $\chi$ ($V$, $L$, and $\sqrt{N}$) is essential for achieving such a universal correlation (see counter examples in Supplementary Fig. 14). Therefore, this trend between $\chi$ and PL symmetry factor suggests that a wholistic consideration and a balance of the intra and interlayer structures is crucial for exciton engineering, which may be tunable through chemical modification of the intra (A-site) and interlayer (LA) cations. Additionally, we tried to assess the capacity of this descriptor in $n = 1$ 2D perovskites, as shown in Supplementary Fig. 16, where the absence of a clear correlation may be attributed to the omission of intralayer cations. Note that various structural descriptors have been proposed for $n = 1$ systems[19–21], which is beyond the scope of this work.

To demonstrate this design principle, we set out to intentionally design a 2D RP perovskite with a small $\chi$ that exhibits high optoelectronic performance. Based on the high-pressure study of $(BA)_2(GA)$

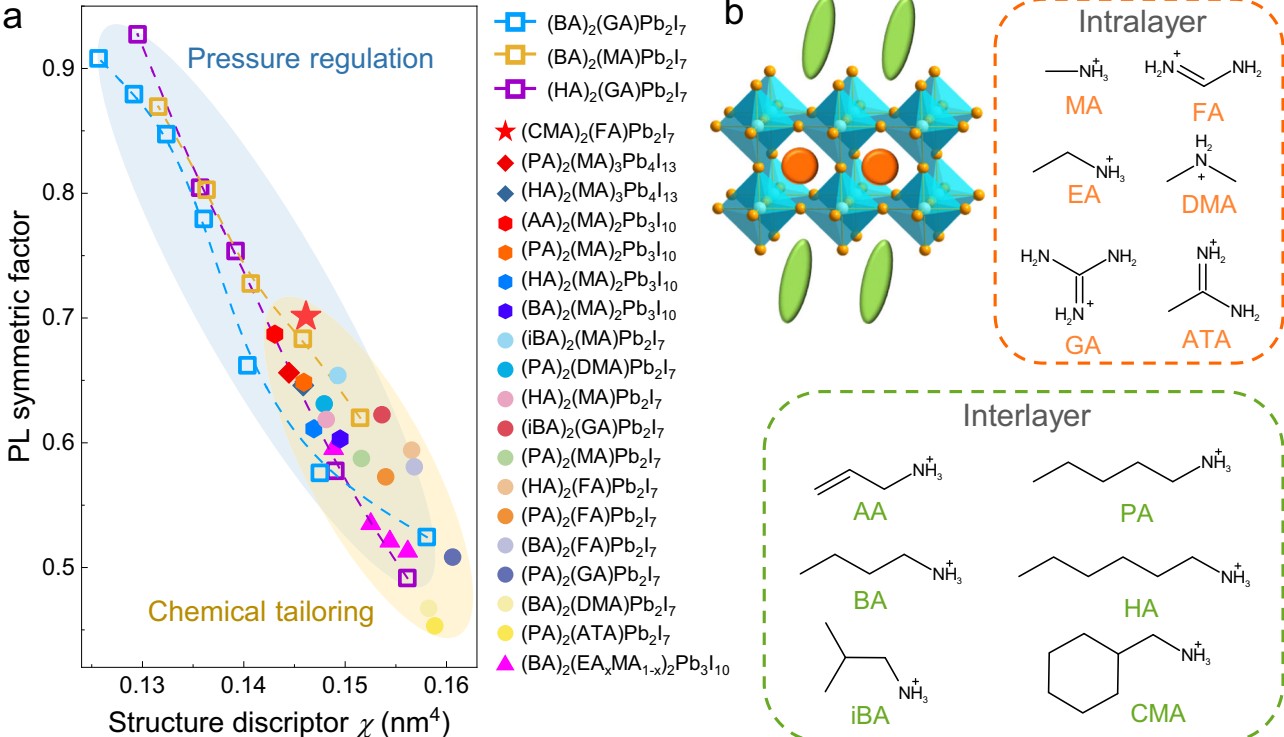

**Fig. 4 | Structure-property relationship revealed from high-pressure studies and survey of many $n > 1$ RP perovskites. a** Symmetric factor as a function of structural descriptor $\chi$ for various 2D RP perovskites under pressure regulation or with chemical tailoring using different LA and A cations and $n$ values. **b** LA and A cations that are incorporated in the intra and interlayer structures of 2D perovskites tabulated in (**a**). Each cation is assigned an abbreviation. Source data are provided as a Source Data file.

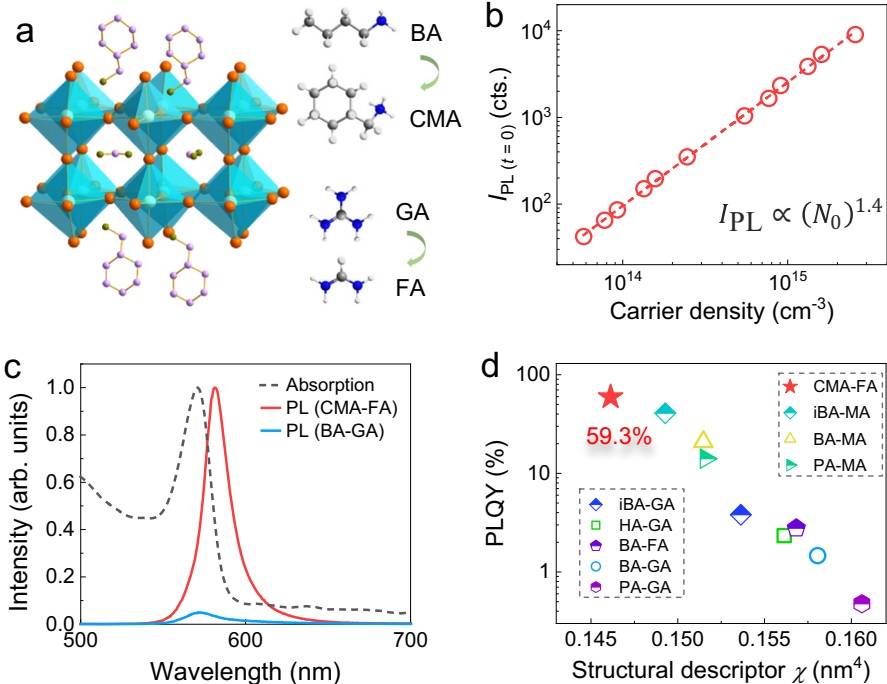

**Fig. 5 | Comprehensive cation engineering to tune the structure and excitonic features. a** Crystal structure of the 2D perovskite compound $(CMA)_2(FA)Pb_2I_7$, which shows the occupation of the perovskite cage by the smaller FA cation (than GA) and the layered structure with more compact interlayer CMA cation (than BA). **b** The exponential value $\alpha$ of the carrier density dependent initial PL intensities for $(CMA)_2(FA)Pb_2I_7$. **c** Absorption and PL spectra of $(CMA)_2(FA)Pb_2I_7$, in comparison with the PL spectrum of $(BA)_2(GA)Pb_2I_7$. The relation between the PL symmetric factor and structural descriptor $\chi$ of $(CMA)_2(FA)Pb_2I_7$ is represented as the red star in Fig. 4a. **d** The relation between the PLQY (in logarithm scale) and structural descriptor $\chi$ for various common $n = 2$ RP perovskites $(LA)_2(A)Pb_2I_7$ (abbreviated as LA-A), among which $(CMA)_2(FA)Pb_2I_7$ has the smallest $\chi$ and exhibits the highest PLQY of 59.3%. Source data are provided as a Source Data file.

$Pb_2I_7$, we sought out a smaller A-site cation that would result in a smaller cage volume and a slightly bulkier, but more rigid LA cation that could pack densely in the interlayer under ambient pressure. We chose the A-site cation formamidinium (FA) and LA cation cyclohexanemethylammonium (CMA)[51], as shown in Fig. 5a, that fit these criterion. We hypothesize that such a combination of intra and interlayer cations (CMA-FA) could achieve a similar structural tuning effect observed for $(BA)_2(GA)Pb_2I_7$ under compression. Since the crystal structure of the resulting compound $(CMA)_2(FA)Pb_2I_7$ has not been previously reported, we determined its crystal structure (Fig. 5a) using single-crystal X-ray crystallography (see details on the synthesis and crystal structure determination in the Supplementary Information and the full crystallographic information in Supplementary Table 2). The structural changes are noticeable when the perovskite cages of $(CMA)_2(FA)Pb_2I_7$ and $(BA)_2(GA)Pb_2I_7$ are overlaid (Supplementary Fig. 17). Notably, as we predicted, the structural descriptor $\chi$ for $(CMA)_2(FA)Pb_2I_7$ is reduced considerably to $0.146\,nm^4$, which is the lowest value among the known $n = 2$ RP perovskites with alkylammonium spacer cations (represented as the red star in Fig. 4a).

To study the excitonic properties and emission performance of $(CMA)_2(FA)Pb_2I_7$, we performed the PL measurements and tracked the behaviors of $I_{PL}$ $(t = 0)$ as a function of carrier density (Fig. 5b and Supplementary Fig. 18). The exponential value $\alpha$ increases from 1.1 for $(BA)_2(GA)Pb_2I_7$ to 1.4 for $(CMA)_2(FA)Pb_2I_7$, indicating the reduction of $E_b$ by the chemical tailoring. Remarkably, $(CMA)_2(FA)Pb_2I_7$ with a smaller $\chi$ value exhibits a more symmetric PL peak and a 21 times higher PL intensity than that of $(BA)_2(GA)Pb_2I_7$ (Fig. 5c). Impressively, the PLQY of $(CMA)_2(FA)Pb_2I_7$ reaches 59.3%, the highest among the known $n = 2$ RP perovskite single crystals (Fig. 5d and Supplementary Fig. 19). The results on the $(CMA)_2(FA)Pb_2I_7$ compound further support the structure-property relationship revealed from high-pressure investigation (Fig. 4a).

## Discussion

We note that the correlation between PLQY[16], which is also reflected in the PL symmetric factor, and the formulated structural descriptor $\chi$ for many families of 2D RP perovskites falls on the same general trend regardless of the exact intra (A) and interlayer (LA) cations (Figs. 4a and 5d). As the structural descriptor $\chi$ decreases, the PL symmetric factor increases, leading to higher PLQY. The presented cases include pressure-manipulated results on $(BA)_2(GA)Pb_2I_7$, $(HA)_2(GA)Pb_2I_7$[40], and $(BA)_2(MA)Pb_2I_7$, as well as numerous chemically tailored 2D perovskites[23,24,38,40,48–50]. Both pressure regulation and chemical tailoring can realize the synergistic manipulation of intra and interlayer structures and enable the effective exciton engineering in 2D RP perovskites for enhanced optoelectronic properties. The schematic illustration of intralayer and interlayer cation substitutions to achieve a synergistic structure and property tuning is shown in Supplementary Fig. 20. These results suggest that there could be ideal LA/A cation pairings that can optimize $\chi$ for the 2D perovskite structure. Selection of a large A-cation, which typically leads to oversized perovskite cage and decreased PL symmetry, may need to be compensated by the selection of an appropriate LA cation that densely packs in a shorter interlayer distance, in order to achieve excellent optoelectronic performance. The structural descriptor $\chi$ introduced in this study, which evaluates the balance between intra and interlayer structural characteristics, is a suitable parameter that can be used for structural design and property prediction of 2D RP perovskites.

We would like to point out that the interlayer interactions between organic spacer cations, including hydrogen bonding[52–54], π-π interactions[16,55], and the formation of intermolecular networks[11,52], could influence the structures and properties of 2D perovskites. All these interactions are relatively strong and complicated, which will bring diverse effects on both structures and properties. Describing

these diverse effects using a single structural parameter is a challenging endeavor, and thus far, the number of cases studied remains limited. Therefore, here we focused on the extensively studied alkyl-lammonium spacer cations without exhibiting such strong interactions to establish a quantitative structure-property relationship. In these cases, the structural descriptor $\chi$ has already included the contribution of the interlayer interactions between spacer cation to some degree, but such contribution is mostly reflected in the geometric consequences after the spacer cations pack into the interlayer space.

In summary, we highlight the importance of understanding the synergistic effects of intra and interlayer structures on the excitonic properties of $n > 1$ 2D RP perovskites. Here we use pressure to tune the 2D perovskite structure and realize controllable exciton engineering from equilibrium of trapped and free excitons to equilibrium of free excitons and carriers. The PL intensity of $(BA)_2(GA)Pb_2I_7$ perovskite shows a 72-fold boost upon compression due to the suppression of exciton trapping, and the photoconductivity exhibits a 10-fold increase because of the decreased exciton binding energy. Importantly, we introduce a general structural descriptor, $\chi$, which considers both intra and interlayer structural characteristics of 2D RP perovskites and quantitatively demonstrate that a decreasing $\chi$ correlates with a more symmetric PL and higher emission intensity. Guided by this principle, we design a $(CMA)_2(FA)Pb_2I_7$ RP perovskite with a smaller intralayer cation (FA) and a more compact interlayer cation (CMA) to emulate the pressure-induced structural changes. This $(CMA)_2(FA)Pb_2I_7$ perovskite with a reduced $\chi$ exhibits an exceptionally high PLQY of 59.3%, the record-high value among known $n = 2$ RP perovskites. Our findings demonstrate a general strategy to engineer excitonic properties of $n > 1$ perovskites based on the synergistic tuning of intra and interlayer structures. The proposed structural descriptor $\chi$ can serve as a guide for the rational design of high-performance 2D perovskites toward efficient optoelectronic devices.

## Methods
### Synthesis of 2D halide perovskites
For synthesizing $(BA)_2(GA)Pb_2I_7$, 5.714 mmol $PbI_2$ and 2.857 mmol guanidine hydrochloride was dissolved in 9 mL HI solution and 1 mL of $H_3PO_2$ in a vial. Then 2.857 mmol butylamine was added into the solution, and all reagents were dissolved under heating and stirring. Then stop heating and red single crystals precipitated during cooling. For synthesizing $(CMA)_2(FA)Pb_2I_7$, 230 mg $PbI_2$ and 172 mg formamidinium iodide were added into 5 mL of HI solution in a vial, then 56 μL cyclohexanemethylamine was added into the solution slowly. The solution was heated at 120 °C and added methanol to produce a clear yellow solution. The vial was then cooled down and left overnight, yielding red single crystals.

### Exfoliating 2D halide perovskites
The 2D perovskite thin films for the absorption measurements were prepared by mechanical exfoliation. Plate-like single crystals were carefully selected and placed on a clear tape. Another clean section of tape was folded over the crystal. A portion of the crystal was then detached for subsequent exfoliation, revealing fresh surfaces, while the remainder stayed on tape. This process was repeated multiple times to yield optically thin 2D perovskite films. The exfoliated 2D perovskite samples were transferred on to a piece of polydimethylsiloxane (PDMS), and then transferred onto the diamond by slow peeling.

### Pressure environment provided by diamond anvil cell
The pressure was controlled by diamond anvil cells (DACs), utilizing ultralow fluorescence diamonds with a culet size of 400 μm. Within the high-pressure sample chamber, constructed from a pre-indented T301 gasket with approximately 50 μm thickness, a laser-drilled hole of approximately 250 μm diameter was positioned at its center. Both the

sample and a ruby ball were loaded into the chamber, and the pressures were determined via the ruby fluorescence method[56]. The ruby fluorescence spectra were fitted using Gauss-Lorentz function for $R_1$ and $R_2$ lines, where the $R_1$ line is used for evaluating the pressure values. The temperature correction of ruby fluorescence is done according to the previous study by Datchi et al.[57]. Detailed pressure calculation is available on https://millenia.cars.aps.anl.gov/gsecars/ruby/ruby.htm. Silicone oil served as the pressure transmitting medium throughout the experiments.

### In situ high-pressure structural characterizations
The in situ single-crystal X-ray diffractions were performed at beamline 13BM-C of GeoSoilEnviroCARS at Advanced Photon Source, Argonne National Laboratory. The wavelength of the monochromatic X-ray beam is 0.434 Å. Structural analysis was carried out using the SHELX software package, coupled with the OLEX2 program[58,59].

### In situ high-pressure optical measurements
All the static measurements were conducted using a home-designed spectroscopy (Gora-UVN-FL, Ideaoptics, China). Absorption measurements utilized a stabilized tungsten-halogen lamp (SLS201L, Thorlabs, USA) as the light source, covering a range of 360–2600 nm. PL measurements were performed using a 405 nm continuous laser for excitation. Laser power adjustment was achieved through a neutral-density filter. The laser beam, focused onto the sample surface by a 20× objective lens (0.42 NA, Mitutoyo, Japan), yielded a spot diameter of 3 μm, with emission collection facilitated by the same objective lens. Temperature-dependent PL data were obtained using a microscope cryostat system (ST-500, Janis, USA) under 405 nm laser excitation. The time-resolved PL spectra were acquired utilizing a custom-built high-pressure transient state spectroscopy system (Timetechspectra, China). A 532 nm picosecond laser, selected from a supercontinuum laser source (SC-OEM, Yangtze Soton Laser, China) via a bandpass filter, operated at a repetition rate of 10 MHz with a pulse width of approximately 100 ps. The time-resolved PL decay kinetics were captured and analyzed employing a time-correlated single photon counting (TCSPC) module (PicoHarp 300) in conjunction with a single photon avalanche diode (SPAD) detector (id100-20, IDQ, Switzerland), featuring an instrument response function of around 180 ps.

### In situ high-pressure photocurrent measurements
A $(BA)_2(GA)Pb_2I_7$ single crystal with a size of about 100 μm × 100 μm × 10 μm was used. To ensure electrical insulation between the electrical leads and the metal gasket, a cubic boron nitride (cBN) layer was interposed between the stainless-steel gasket and the diamond culet. Current-time data were recorded using a Keithley 6517B high-resistance measurement system, with a applied bias voltage of 10 V. Illumination was provided by a LED light source emitting at 500 nm, generating an irradiation intensity of approximately 10 W cm$^{-2}$ on the sample.

### In-lab single crystal X-ray diffraction study
A red crystal of $(CMA)_2(FA)Pb_2I_7$ with approximate size of $0.04 \times 0.04 \times 0.01$ mm$^3$ was selected and attached to the tip of a MiTeGen (MicroMount). Crystal evaluation and data collection were conducted on a Bruker D8 VENTURE PhotonIII four-circle diffractometer with Cu $K\alpha$ ($\lambda = 1.54178$ Å) radiation.

### PL quantum yield measurement
The PLQY of 2D perovskite single crystals were measured using the following equation[60–62]:

$$\phi_{sample} = \phi_{reference} \frac{I_{sample} \times A_{reference}}{I_{reference} \times A_{sample}} \tag{4}$$

where $\phi$, $I$, $A$ are PLQY, PL intensity and absorption of the sample and reference. The reference sample, tris(8-hydroxyquinoline)aluminum (Alq$_3$), was fabricated by spin-casting a Alq$_3$-doped polymethylmethacrylate (PMMA) film onto a glass slide. The precursor solution was prepared by blending Alq$_3$ (8 mg mL$^{-1}$, 1 mL, dissolved in dichloromethane) with PMMA (100 mg mL$^{-1}$, 1 mL, dissolved in chlorobenzene). Subsequently, 200 μL of the precursor solution was spin-coated onto a 1 × 1 inch glass substrate at 3000 rpm for 30 seconds. The PLQY of the Alq$_3$ reference sample was calibrated to be 32.4% under 405 nm excitation using a Hamamatsu C9920-02 PL quantum yield measurement system with an integrating sphere.

## First-principles calculations

First-principles calculations were conducted within the framework of density functional theory, utilizing the Vienna ab initio simulation package (VASP, ver. 5.4.4)[63]. The projected-augmented wave potentials employed, accounting for 14 valence electrons for Pb (5d$^{10}$6s$^2$6p$^2$), 7 for I (5s$^2$5p$^5$), 4 for C atoms (2s$^2$2p$^2$), 5 for N atoms (2s$^2$2p$^3$) and 1 for H (1s$^1$). The exchange-correlation energy was parameterizd by the generalized gradient approximation of Perdew-Burke-Ernzerhof optimizd for solid[64,65]. The Brillouin zone sampling was achieved through a k-points grid mesh, ensuring the smallest allowed spacing between each k-point of 0.2 Å$^{-1}$. Spin-orbit coupling effects were considered, potentially influencing the band structure of the system[66]. Structural optimization, encompassing atomic positions, cell variables, and volume, was carried out at the target pressure until interatomic forces fell below 0.01 eV Å$^{-1}$. Subsequently, the dielectric and piezoelectric tensors were determined using density functional perturbation theory[67].

## Data availability

The X-ray crystallographic structures for (CMA)$_2$(FA)Pb$_2$I$_7$ have been deposited at the Cambridge Crystallographic Data Centre (CCDC), under deposition number 2320590. These data can be obtained free of charge from The Cambridge Crystallographic Data Centre via www.ccdc.cam.ac.uk/data_request/cif. The data supporting the key findings of this study are available within the article and its Supplementary Information files. Any further relevant data are available from the corresponding authors on request. Source data are provided with this paper.

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

## Acknowledgements

This work is supported by the National Nature Science Foundation of China (NSFC) (Grant Nos. 22275004, U1930401, and 17N1051-0213), Shanghai Science and Technology Committee (No. 22JC1410300), and Shanghai Key Laboratory of Novel Extreme Condition Materials (No. 22dz2260800). W.M.K. thanks the National Science Foundation (NSF) Graduate Research Fellowship for support. M.P.H. and S.J. thank the support by US Department of Energy, Office of Basic Energy Sciences, Division of Materials Sciences and Engineering, under Award DE-SC0002162. Some experiments are supported by the Synergic Extreme Condition User Facility. Portions of this work were performed at Geo-SoilEnviroCARS (The University of Chicago, Sector 13), Advanced Photon Source (APS), Argonne National Laboratory. GeoSoilEnviroCARS is supported by the National Science Foundation – Earth Sciences (EAR –1634415). This research used resources of the Advanced Photon Source, a U.S. Department of Energy (DOE) Office of Science User Facility operated for the DOE Office of Science by Argonne National Laboratory under Contract No. DE-AC02-06CH11357.

## Author contributions

S.G., Y.F., S.J. and X-J.L. conceived the project. W. M-K., X-Y. L., H.H., J.G. and M.P.H., synthesized single-crystal samples. S.G., Y.M., Y.Z. and H.L. collected the photoluminesence, absorption, and photocurrent data. K.B. and Q.H. carried out the first-principle calculations. S.G. and K.B. collected and analyzed the high-pressure single crystal XRD data with the assistance of D.Z. S.G. and D.W. performed the in situ low temperature PL measurement. S.G., W. M-K., X-J. L., Y.F. and S.J. wrote the manuscript and revised by Q.H., Y.D. and W.Y. All authors have interpreted the findings, commented on the paper, and approved the final version.

## Competing interests

The authors declare no competing interests.
