## [Peer Review File · Nature Communications]

Exciton engineering of 2D Ruddlesden–Popper perovskites by synergistically tuning the intra and interlayer structureReviewers' comments:

Reviewer #1 (Remarks to the Author):

In this manuscript, the authors have tuned the excitonic and absorption properties of (BA)₂(GA)Pb₂I₇. The insights further helped to formulate a descriptor for understanding structure - exciton property relationship. Though the study has been conducted carefully, I found several key issues that preclude me from recommending it for publication.

1. The primary concern is the novelty aspect of this work. There are several reports on the impact of pressure on the excitonic and emission properties of 2D-layered materials (Proc. Natl. Acad. Sci. U. S. A. 2018, 115, 8076–8081, check review J. Phys. Chem. Lett. 2020, 11, 12, 4693 for numerous examples). I understand that the exact combination of A and spacer cation has not been explored probably yet. However, I don't see any drastic difference in the overall pressure-induced effects between the current and previously published reports. Thus, the present work strongly lack the novelty aspect.
2. It is unclear what is precisely unique about the BA-GA combination at the atomistic level. The authors formulate some empirical relationships later in the manuscript; however, the atomistic details of the structure-pressure-electronic-excitonic properties relationships are mainly missing.
3. It's common to find multiple phase transitions in layered halide perovskites under pressure. However, here authors do not find any such transitions. There must be an elaborate discussion explaining such a phenomenon.
4. The unified structural descriptor χ is quite ad-hoc and has been tested on an extremely small number of systems. Furthermore, the authors have tested this for n=2 layered halide perovskites only, reducing its applicability for other 2D perovskites. I firmly believe that stating some formulas as "universal" or "unified" needs rigorous testing with large datasets. The current manuscript clearly fails to do so.
5. There are many discussions on pressure-induced band narrowing and consequent shallow-defect formation, which can substantially effect the emission properties. However, it's surprising that the authors neglected such an essential effect in their study.
6. The computational study is quite substandard. The band dispersion strongly depends on spin-orbit coupling as these materials have heavy elements. However, authors do not consider such methods.
7. The band structure in Figure 3d exhibits an indirect bandgap at high pressure. The authors entirely ignore such results and their probable implications on the optoelectronics.
8. The Wannier exciton model for 2D layered perovskites may not be suitable due to presence of strongly bounded excitons. Authors should be careful using such model for their work.

Reviewer #2 (Remarks to the Author):

The authors report on exciton engineering in Ruddlesden-Popper layered metal halide

perovskites by manipulating intra- and inter-layer structures. Specifically, the applied pressure to regulate the anisotropic structure and modulate the excitonic characteristics in n-butylammonium (BA) / guanidinium (GA) lead iodide-based systems of $n = 2$ composition ($\text{BA}_2\text{GA}_2\text{Pb}_2\text{I}_7$) and introduced a descriptor relating the structural characteristics and the photoluminescence quantum yield (PLQY). They relied on the descriptor to identify a new cyclohexanemethylammonium and formamidinium-based layered perovskite with exceptional PLQY. This study reveals valuable insights into the molecular engineering of layered (2D) perovskites that is of broad scientific interest. However, I cannot recommend considering it for publication before addressing the following critical remarks.

- The authors introduce a “quantitative” descriptor that correlates the intra- and inter-layer structural characteristics to the excitonic properties of 2D perovskites. However, the “inter-” and “intra-layer” characteristics are not well defined. Moreover, based on the Figure 1 schematics, the authors exclude the organic intra-layer interactions, and it remains unclear how pressure as an external stimulus could be used to control these factors to correlate these properties to the excitonic features. In general, the description of the structural changes needs to be carefully reconsidered and clarified upon revision.
- The authors further suggest the generality of the approach, yet they explore a relatively limited scope of 2D perovskites that have not been assessed systematically by any outlined rationale. Although they summarize some of the materials considered (e.g., Figure 4a) later in the manuscript, there is no rationale behind the selection, which is relevant earlier in the discussion. Further clarification of the design principles for the systems is required, and, in particular, the rationale behind starting with the BA-GA system remains entirely unclear and needs to be elaborated on, as well as the focus on $n = 2$ compositions (e.g., Why this particular organic spacer and A cation? How does this help proceed with the objectives to clarify the structure-property relationships and the role of the inter/intra-layer interactions, particularly considering the relatively uncommon GA-based A-cation? What stimulates the $n = 2$ compositional focus, and would the approach apply to $n = 1$ compositions?). In addition, the pressure levels used need to be put into context and related to the specific design for a better understanding (e.g., they start with a 2.1 GPa PL investigation without any rationale provided to follow the approach).
- In addition, the “synergistic effects” of intra- and inter-layer structures are emphasized to justify the approach, which is generally applicable beyond this particular study, as overall structural changes in layered perovskite materials result from the interrelated structural changes across layers of the hybrid perovskite structure. How is this distinctly considered within this study, and what justifies the emphasis on “synergistic effects” in this context?
- In defining the structural descriptor, the authors also consider “non-hydrogen atoms”, which should be carefully clarified in the main manuscript. In addition, they identify volume and interlayer distance as critical parameters for establishing the “correlation” with optoelectronic properties. How does this correlation apply to the simple lattice parameters (e.g., d -values) already representative of the overall structure? Why is pressure-induced change critical; are irreversible structural changes considered?
- The literature is mostly comprehensive, yet the authors must put previous reports on pressure-induced transformations of layered hybrid perovskites into context. Similarly, they should refer to prior reports using PLQY as a predictive material design parameter.

In summary, this is a very interesting and insightful analysis that provides important insights into the design of 2D perovskites. However, the approach requires more clarity and putting it into context concerning prior work on the topic for a better understanding of the scope and impact. Finally, the authors should carefully address the design principles behind the model systems used for this purpose and the role of pressure in controlling the changes. I

appreciate your consideration.

Reviewer #3 (Remarks to the Author):

“Exciton engineering of 2D Ruddlesden-Popper perovskites by synergistically manipulating intra- and inter-layer structures” by S. Guo demonstrates that compressive strain can significantly modulate the optoelectronic properties in 2D perovskites. In particular, they observe a large increase in the brightness of PL, a change in shape (more symmetric) of the emission spectrum, and a red-shift as pressure is applied. This is coupled also with pressure-dependent changes in the absorbance profile, and photoconductivity of the 2D materials, which the authors ascribe to reductions in exciton binding energy pushing the material from mostly bound excitons, toward mostly free excitons, and eventually free charges. Based on these results, and on simulations which show how the compressibility of 2D perovskites is affected by the A-site cation and spacer cation, the authors choose a notoriously small A-site cation (FA, of FAPbI₃ perovskite fame) and a rigid spacer cation (CMA) to minimize a structural descriptor X they have developed. This new FA-CMA 2D R-P perovskite is structurally more similar to the strained perovskites they performed most of their characterization on, and as a result its emission (i.e., PLQY) is very bright.

The experimental work and presentation are very clear and high-quality, though the concepts are not entirely novel as several other works have performed optical characterization of strained 2D perovskites: [10.1039/D2NR06816H](https://doi.org/10.1039/D2NR06816H), [10.1021/acscenergylett.7b00284](https://doi.org/10.1021/acscenergylett.7b00284), and the authors themselves have published adjacent papers in this space: [10.1126/sciadv.add1984](https://doi.org/10.1126/sciadv.add1984). Still, I feel that the work is high quality, adds considerably to scientific understanding, and as such should be published.

My one question I hope the authors consider in a minor revision is whether or not the mechanism behind the increased PL can be better understood—perhaps through calculation results that already exist. It strikes me that for PL to go up so strongly, simply freeing the exciton, or delocalizing it, can actually produce the opposite of the observed effect since this will allow excitons to diffuse further until they reach a trap site or annihilate with one another. Since PL is the result of radiative transitions winning a competition between themselves, non-radiative recombination, and dissociation, what I suspect is happening is that the oscillator strength of the exciton emission is being modulated significantly (in addition to its binding energy changing) and thus the radiation is becoming faster and competing better with non-radiative recombination which increases the brightness. This likely occurs because the compression (and indeed using FA-CMA) push the exciton hole and electron density closer and provide more spatial overlap for optical transitions to occur.

Point-by-point Responses to the Reviewers' Comments

(Manuscript ID: NCOMMS-23-34853)

We are very grateful to the reviewers for their time and efforts to review our manuscript. Their comments are very helpful for further improving the quality of our manuscript and we address reviewers' concerns thoroughly below.

Reviewer #1

In this manuscript, the authors have tuned the excitonic and emission properties of (BA)₂(GA)Pb₂I₇. The insights further helped to formulate a descriptor for understanding structure – exciton property relationship. Though the study has been conducted carefully, I found several key issues that preclude me from recommending it for publication.

Response: We sincerely appreciate the reviewer's positive comments such as "the study has been conducted carefully" and "insights further helped...". We would like to note that we further presented a new 2D perovskite of (CMA)₂(FA)Pb₂I₇ with excellent optoelectronic properties and now further added many new examples to further support the generality of the proposed structure-property correlation, as discussed below in the point-by-point responses to the comments.

Comment 1: *The primary concern is the novelty aspect of this work. There are several reports on the impact of pressure on the excitonic and emission properties of 2D-layered materials (Proc. Natl. Acad. Sci. U. S. A. 2018, 115, 8076–8081, check review J. Phys. Chem. Lett. 2020, 11, 12, 4693 for numerous examples). I understand that the exact combination of A and spacer cation has not been explored probably yet. However, I don't see any drastic difference in the overall pressure-induced effects between the current and previously published reports. Thus, the present work strongly lacks the novelty aspect.*

Response: Although several previous studies have reported the impact of pressure on the excitonic and emission properties of 2D RP perovskites, none of them have studied different A cations in perovskite cages. Furthermore, no previous studies have addressed the question of how the intra- and interlayer cations interact synergistically and influence the perovskite structure and properties under applied pressure. For the mentioned two references, the PNAS paper didn't investigate the emission properties of 2D RP perovskites; while the JPCL review paper or other literature focus on $n = 1$ 2D RP perovskites where only the interlayer spacer (LA) cation exists (with no A cations).¹⁻⁶ For the more tunable and structurally complex $n \geq 2$ 2D perovskites, no research to-date has considered the structural modulation and design of

2D perovskites from both the inter- and intralayer structural perspective (*i.e.* both LA and A cations) to gain a more complete understanding and to develop high-performance materials.

The novelty and contrast between our study and previous reports include:

- 1) We effectively and continuously modulate the excitonic properties, including both exciton-phonon coupling and exciton binding energy, and comprehensively characterize the photophysical processes by *in situ* steady-state, time-resolved, and temperature-dependent PL spectroscopies as well as absorption spectroscopy.
- 2) We propose a new structural descriptor χ that considers both the intra- and interlayer structural characteristics in the structurally more tunable $n \geq 2$ 2D perovskites, and establish a quantitative relationship between χ and PL property - including 39 data points from 2D perovskite compounds.
- 3) We further translate the knowledge gained from high-pressure study to new material design, and develop a brand-new compound, $(\text{CMA})_2(\text{FA})\text{Pb}_2\text{I}_7$, which possesses an optimized χ and exhibits an impressive PLQY of 59.3%, the record-high value among known $n = 2$ RP perovskites.

Therefore, our findings unambiguously elucidate the intricate structure-property relationships in ***$n \geq 2$ 2D perovskites with different LA and A cations***, offering novel insights into the structural design of 2D perovskites. More broadly, our work represents ***a new paradigm of research*** in which the high-pressure studies are integrated with innovative materials design and discovery in a seamless fashion.

Comment 2: *It is unclear what is precisely unique about the BA-GA combination at the atomistic level. The authors formulate some empirical relationships later in the manuscript; however, the atomistic details of the structure-pressure-electronic-excitonic properties relationships are mainly missing.*

Response: We thank the reviewer for this comment. The incorporation of the over-sized GA^+ cation in the “model” 2D BA-GA perovskite expands the perovskite cage, resulting in a notably large χ value of 0.158 nm⁴ at ambient conditions (see Figure 4a). This leaves ample room for pressure tuning of the structural descriptor χ , thereby enabling a more wholistic investigation of structure-property relationship across a broader structural range.

To clarify this point, the following discussion has been added in the Revised Manuscript on Page 5: “To uncover the intra and interlayer structural features that govern the excitonic properties, we start by examining the effect of pressure modulation on photoluminescence and photoconductivity of the 2D RP perovskite $(\text{BA})_2(\text{GA})\text{Pb}_2\text{I}_7$. We choose this perovskite because it incorporates an oversized A-cation (GA) into the perovskite cage, resulting in considerable octahedral distortion and a high level of exciton trapping and asymmetric PL peak. Additionally, the LA cations (BA) are flexible alkyl chains that make the interlayer is highly

compressible. Therefore, $(\text{BA})_2(\text{GA})\text{Pb}_2\text{I}_7$ is an ideal perovskite to investigate whether pressure-induced structural changes can improve excitonic behavior and analyze the nuanced structural-property relationships across a vast parameter space.”

The description of the atomistic details of the structure-pressure-electronic-excitonic properties relationships has been included on Page 11 of the Revised Manuscript: “Therefore, the pressure-induced structural modulation results in the suppression of trapped excitons and promotes the further dissociation of free excitons, which enhances PL and photoconductivity performance. Such structural change is a combination of intra and interlayer effects. Within the intralayer, there is a decrease in the Pb-I bond length and an increase in the Pb-I-Pb bond angle under compression, resulting in a reduction in the perovskite cage volume and the idealization of the perovskite framework. Consequently, the bandgap narrows due to the increased overlap between Pb(6s) and I(5p) orbitals. In addition, these structural variations suppress exciton trapping through phonon hardening^{24,43}. Furthermore, interlayer distance decreases upon compression which leads to increased “packing density” of spacer cations and the weakening of the dielectric confinement and consequently reduces the exciton binding energy^{2,50}.”

Comment 3: *It's common to find multiple phase transitions in layered halide perovskites under pressure. However, here authors do not find any such transitions. There must be an elaborate discussion explaining such a phenomenon.*

Response: We would like to clarify that many 2D halide perovskites do not exhibit phase transitions under high pressure, such as $(\text{PEA})_2\text{PbI}_4$, $(\text{BA})_2(\text{MA})\text{Pb}_2\text{I}_7$, and $(\text{HA})_2(\text{GA})\text{Pb}_2\text{I}_7$ ^{3, 5, 7, 11-16}. Therefore, the observation herein is hardly unusual. In our study, we conducted *in situ* single-crystal XRD to precisely determine the high-pressure crystal structure. As illustrated in Figure R1, no discernible splitting or appearance of new diffraction spots exist, signifying the absence of phase transition up to a certain pressure (*i.e.* 5.5 GPa). Certainly, pressure-induced phase transition or disordering may happen when pressure reaches a critical value, but it is out of the scope of this work.

Figure R1. *In situ* single-crystal X-ray diffraction images of $(\text{BA})_2(\text{GA})\text{Pb}_2\text{I}_7$ at 0.4 GPa and 5.5 GPa, where no phase transition can be observed.

Comment 4: *The unified structural descriptor χ is quite ad-hoc and has been tested on an extremely small number of systems. Furthermore, the authors have tested this for $n=2$ layered halide perovskites only, reducing its applicability for other 2D perovskites. I firmly believe that stating some formulas as "universal" or "unified" needs rigorous testing with large datasets. The current manuscript clearly fails to do so.*

Response: Thank you for this comment. We came up with this expression of the structural factor after empirically testing many other possible combinations of different structural parameters which did not show a strong correlation. The step-by-step analysis for arriving at the structure descriptor χ is shown in Supplementary Figures 14 and 15, along with detailed description presented in the Revised Supplementary Information. The proposed structural descriptor χ comprehensively considers the intra and interlayer structural features and is not ad hoc at all, once we understand the key idea that the balance between the LA cation and A cation for the $n \geq 2$ RP perovskites is critical for their excitonic properties.

Supplementary Fig. 14 PL symmetric factor as a function of (a) average Pb-I bond length, (b) cage volume V , (c) interlayer distance, (d) packing density, and (e) structure descriptor χ for various $n \geq 2$ 2D RP perovskites (listed in the legends). The correlation coefficients are given for each plot. Comprehensive consideration of both the intralayer (V) and interlayer (L and \sqrt{N}) structural characteristics in the proposed structure descriptor χ is essential for achieving a good correlation relationship.

Supplementary Fig. 15 The correlation coefficient between PL symmetric factor and packing density as a function of the exponential value α of N , where the strongest correlation can be found when the α near -0.5 .

The following discussion has been added on Page 8 of the Revised Supplementary Information: “The correlation between PL symmetric factor and various structural parameters can be estimated by correlation coefficient. The correlation coefficient $\rho_{X,Y}$ between two random variables X and Y with expected values \bar{X} and \bar{Y} and standard deviations σ_X and σ_Y is defined as:

$$\rho_{X,Y} = \frac{cov(X,Y)}{\sigma_X\sigma_Y} = \frac{E[(X - \bar{X})(Y - \bar{Y})]}{\sigma_X\sigma_Y}$$

where E is the expected value operator, cov means covariance. The value of a correlation coefficient ranges between -1 and $+1$. The correlation coefficient is $+1$ in the case of a perfect direct (increasing) linear relationship (correlation), -1 in the case of a perfect inverse (decreasing) linear relationship (anti-correlation). As it approaches zero there is less of a relationship (closer to uncorrelated). The closer the coefficient is to either -1 or 1 , the stronger the correlation between the variables.

Firstly, in terms of intralayer structure, compression induces a decrease in the Pb-I bond length and an increase in the Pb-I-Pb bond angle. We plot the PL symmetric factor as a function of average Pb-I bond length, only getting a correlation coefficient $\rho = -0.789$ (Supplementary Figure 14a). Compression also leads to an increase in the Pb-I-Pb bond angle, thus resulting in an idealization of the perovskite cage and a subsequent reduction in its cage volume (V). The correlation coefficient improves to -0.904 when we use cage volume as the structural descriptor (Supplementary Figure 14b).

Then, in terms of interlayer structure, compression induces a decrease in the interlayer distances. However, no clear correlation can be found between PL symmetric factor and interlayer distances (Supplementary Figure 14c). The probable reason is because interlayer

distance only depicts the crystal structure and do not consider the dielectric constant of the spacer cations. For example, HA and BA fall on different trendlines because HA has two extra C atoms and causes a larger interlayer distance, but their dielectric constants are very similar. Therefore, we choose “packing density” of the spacer cations instead of simple interlayer distance, which essentially convey how well the spacer cations pack and are normalized from interlayer distance as follows:

$$\text{Packing density} = L \times N^\alpha$$

where the L and N are interlayer distance and number of non-hydrogen atoms in the spacer cation, respectively. The correlation coefficient between PL symmetric factor and packing density as a function of exponential value α is shown in Supplementary Figure 15, where the strongest correlation can be found when the α near -0.5 . Thus, the packing density in this case can be defined as $\frac{L}{\sqrt{N}}$, which shows a much better correlation coefficient $\rho = -0.686$ with PL symmetric factor (Supplementary Figure 14d).

A further improved correlation coefficient can be achieved when we synergistically considered both the intra- and interlayer structural parameters, obtaining a new comprehensive structural descriptor χ :

$$\chi = \text{Cage volume} \times \text{Packing density} = V \times \frac{L}{\sqrt{N}}$$

where V and L refer to the perovskite cage volume and interlayer distance, respectively, and N is the number of non-hydrogen atoms in the interlayer spacer cation. The correlation coefficient between PL symmetric factor and this structural descriptor χ reaches -0.961 (Figure S14e), which indicates a very good correlation. The step-by-step correlation process above also confirms that each component of χ (V , L , and \sqrt{N}) is essential for achieving such a universal correlation.”

In order to broaden the generality and universality of this structural factor, we carefully surveyed the literature and have added a broad range of reported 2D perovskites with various aliphatic LA and A cations into our structural-property relationships, and further extended this principle to $n = 3$ and $n = 4$ 2D halide perovskites. The comparison between the previous version and the updated Figure 4a is depicted in Figure R2, where the newly added 2D perovskites are highlighted in red. The structural descriptor χ consistently demonstrates a strong correlation with the symmetric factor of PL peaks across an extensive set of a total 39 data points of 2D perovskites incorporating 6 different aliphatic LA spacers and 6 different A cations. Therefore, we are confident that the descriptor χ serves as an effective and universally applicable predictive parameter for the structural design of high-performance 2D perovskites. For the sake of accuracy, we have replaced the term “unified” to “new” in the Revised Manuscript.

Figure R2. (a) The previous version of Figure 4a, which shows the PL symmetric factor as a function of structural descriptor χ for various $n = 2$ RP perovskites under pressure regulation (BA-GA, HA-GA, and BA-MA) or with chemical tailoring [(LA)₂APb₂I₇, LA = CMA, HA, PA, BA, and iBA; A = GA, FA, and MA]. (b) The updated version of Figure 4a, where a large scope of 2D perovskites with various LA and A cations have been added into this structural-property relationship, and the principle has been successfully expanded to $n = 3$ and $n = 4$ 2D halide perovskites. The new added 2D perovskites are highlighted in red.

Comment 5: There are many discussions on pressure-induced band narrowing and consequent shallow-defect formation, which can substantially effect the emission properties. However, it's surprising that the authors neglected such an essential effect in their study.

Response: We totally agree with the reviewer that defects would influences the emission properties. To elucidate this, we first performed the temperature-dependent PL measurement on (BA)₂(GA)Pb₂I₇ at ambient conditions. As shown in Figure R3a, no emission from defects can be observed. The low-energy tail of the PL peak is attributed to exciton-trapping.

We further measured the relative PLQY as a function of carrier density for (BA)₂(GA)Pb₂I₇ at different pressures, as shown in Figure R3b. At ambient conditions, the nearly constant value can be observed at carrier density below 10¹⁶ cm⁻³, validating that the luminescence in (BA)₂(GA)Pb₂I₇ is from excitonic recombination rather than defect-assisted recombination. Upon compression, the PLQY of (BA)₂(GA)Pb₂I₇ starts to rise with increasing injected carrier density due to saturation of the defects. This is because excitons are more easily to be captured by defects when they become delocalized under compression.

Figure R3. (a) Temperature-dependent PL spectra of $(\text{BA})_2(\text{GA})\text{Pb}_2\text{I}_7$ at ambient conditions, where the emission from defects can hardly be found. (b) Relative PL quantum yield (PLQY) for $(\text{BA})_2(\text{GA})\text{Pb}_2\text{I}_7$ as a function of carrier density at different pressures.

To avoid the influence of defects, we have chosen to use the symmetric factor of the PL peak rather than the PL intensity in our structure-properties relationship (Figure 4a). This selection allows for a more equitable comparison of the PL properties among different 2D perovskites.

The corresponding description has been added on Page 9 of the Revised Manuscript: “It is worth noting that the PLQY of $(\text{BA})_2(\text{GA})\text{Pb}_2\text{I}_7$ starts to increase as the injected carrier density rises at high pressure due to saturation of defects. This phenomenon occurs because excitons tend to be more readily captured by defects when they become delocalized under compression. This behavior indicates the transition of main carrier species from the equilibrium between trapped and free excitons to the equilibrium between free excitons and free carriers through the reduction of E_b ¹⁷⁻¹⁸.”

Comments 6&7: *The computational study is quite substandard. The band dispersion strongly depends on spin-orbit coupling as these materials have heavy elements. However, authors do not consider such methods. The band structure in Figure 3d exhibits an indirect bandgap at high pressure. The authors entirely ignore such results and their probable implications on the optoelectronics.*

Response: We thank the reviewer for this comment. We would like to respond to Comments 6 and 7 collectively, as both concern about the theoretical calculation. We agree with the reviewer that the contribution of spin-orbit coupling is important for these materials containing heavy elements like Pb. In the revised version, we have recalculated the band structures by taking into account the effects of spin-orbiting coupling. As shown in Figure R4a and Figure S12, the new results show a direct bandgap for $(\text{BA})_2(\text{GA})\text{Pb}_2\text{I}_7$ at each pressure.

Upon compression, the enhanced dispersion observed near both the CBM and VBM suggests a decrease in effective mass and an increase in carrier mobility. Subsequently, we recalculated the exciton-binding energy E_b utilizing these updated band structures, as shown in Figure R4b. The specific values of E_b changed somewhat but the general trend and conclusion did not change. Figure 3d and 3e have been appropriately revised, reflecting these modifications, in the Revised Manuscript.

Figure R4. (a) Calculated electronic structures of $(\text{BA})_2(\text{GA})\text{Pb}_2\text{I}_7$ at 0.4 and 2.9 GPa, which reveal a more dispersive nature near the band edges and suggest an enhanced carrier mobility at a higher pressure. (b) Calculated exciton binding energy E_b due to structural evolution as a function of pressure.

Comment 8: *The Wannier exciton model for 2D layered perovskites may not be suitable due to presence of strongly bounded excitons. Authors should be careful using such model for their work.*

Response: As we know, excitons can be grouped into two categories: Frenkel excitons and Wannier-Mott excitons¹⁹. Frenkel excitons are tightly bound excitons which have a radius smaller than the crystal lattice constant. These are often found in insulators or organic semiconductors, as they are often bound to specific atoms or molecules. On the other hand, Wannier-Mott excitons have a larger excitonic radius, which are observed in quantum dots, single-walled carbon nanotubes, 2D transition metal dichalcogenides (TMDs), and organic-inorganic halide perovskites. In the cases of $n \geq 2$ 2D halide perovskites, the Bohr radius of excitons generally exceeds 2 nm, considerably larger than the in-plane lattice constants which are less than 1 nm. Consequently, Wannier-Mott exciton model proves to be more appropriate for the 2D halide perovskites. As a result, this model has been extensively applied in the literature to characterize the exciton behavior in 2D perovskites²⁰⁻²³.

Reviewer #2

*The authors report on exciton engineering in Ruddlesden-Popper layered metal halide perovskites by manipulating intra- and inter-layer structures. Specifically, the applied pressure to regulate the anisotropic structure and modulate the excitonic characteristics in *n*-butylammonium (BA) / guanidinium (GA) lead iodide-based systems of *n* = 2 composition (BA₂GA₂Pb₂I₇) and introduced a descriptor relating the structural characteristics and the photoluminescence quantum yield (PLQY). They relied on the descriptor to identify a new cyclohexanemethylammonium and formamidinium-based layered perovskite with exceptional PLQY. This study reveals valuable insights into the molecular engineering of layered (2D) perovskites that is of broad scientific interest. However, I cannot recommend considering it for publication before addressing the following critical remarks.*

Response: We sincerely appreciate the reviewer's positive comments. The followings are the point-by-point responses to each of the comments.

Comment 1: *The authors introduce a “quantitative” descriptor that correlates the intra- and inter-layer structural characteristics to the excitonic properties of 2D perovskites. However, the “inter-” and “intra-layer” characteristics are not well defined. Moreover, based on the Figure 1 schematics, the authors exclude the organic intra-layer interactions, and it remains unclear how pressure as an external stimulus could be used to control these factors to correlate these properties to the excitonic features. In general, the description of the structural changes needs to be carefully reconsidered and clarified upon revision.*

Response: We thank the reviewer for the comment and suggestion. The “inter-” and “intra-layer” characteristics have been well defined in the revised manuscript (on Page 9). Briefly, in the model *n* = 2 perovskite compound BA₂(GA)Pb₂I₇, the inorganic sublattice, composed of two-layers of corner-shared [PbI₆]⁴⁻ octahedra (Figure R5a top), are separated by the organic BA cations (Figure R5a bottom). Subsequently, the band structure of these 2D perovskites along the stacking direction is aligned as type-I where the CBM and VBM are determined by the inorganic perovskite layer (Figure R5b). In such a configuration, carriers are confined in a multiple-quantum-well system due to quantum and dielectric confinement effects. Here, we designate the layers that containing [PbI₆]⁴⁻ octahedra (potential well) and BA cations (dielectric barrier) are the intra- and interlayer structures, respectively. In the structural descriptor χ , we use the cage volume (*V*) and packing density of spacer cation (L/\sqrt{N} , interlayer distance normalized by the square root of the number of non-hydrogen atoms) to quantitatively describe the intra- and interlayer structure characteristics of 2D perovskites, respectively. A detailed explanation of how cage volume and interlayer distance are

determined can be found on Page 8 of the Supplementary Information.

Figure R5. (a) Crystal structure of $(\text{BA})_2(\text{GA})\text{Pb}_2\text{I}_7$, which shows the occupation of the GA cation in the perovskite cage formed by eight $[\text{PbI}_6]^{4-}$ octahedra (top panel) and the layered structure with interlayer BA cations (bottom panel). The perovskite cage volume and interlayer distance are also schematically illustrated. This figure is also displayed as Figure 3a in the Revised Manuscript. (b) Energy level alignment (top) and dielectric constant profile (bottom) along the stacking direction of 2D halide perovskite²⁴.

For the influence of organic intralayer interactions, we would like to note that owing to their spatial separation by the Pb-I octahedra (*i.e.* enclosed in the perovskite cages), the effects of interactions between these cations are minimal²⁵. The contribution of the intralayer cations to the structure can be attributed to the organic-inorganic interactions, such as those causing structural distortion of the inorganic Pb-I framework. In our study, the incorporation of a large intralayer cation GA leads to the expansion of perovskite cage and induces significant octahedral distortion of the inorganic framework. Importantly, the expanded cage of the GA perovskite, as captured by the structure parameter of cage volume (V), setting it apart from other 2D perovskites, offers an unprecedented entry into a hitherto unexplored structural region. This is particularly apparent when applying external high pressure, which permits larger tunability for revealing the structure-property relationship.

We have added more discussion about the structural changes and their correlation with the excitonic features on Page 11 of the Revised Manuscript: “Therefore, the pressure-induced structural modulation results in the suppression of trapped excitons and promotes the further dissociation of free excitons, which enhances PL and photoconductivity performance. Such structural change is a combination of intra and interlayer effects. Within the intralayer, there is a decrease in the Pb-I bond length and an increase in the Pb-I-Pb bond

angle under compression, resulting in a reduction in the perovskite cage volume and the idealization of the perovskite framework. Consequently, the bandgap narrows due to the increased overlap between Pb(6s) and I(5p) orbitals. In addition, these structural variations suppress exciton trapping through phonon hardening^{24,43}. Furthermore, interlayer distance decreases upon compression which leads to increased “packing density” of spacer cations and the weakening of the dielectric confinement and consequently reduces the exciton binding energy^{2,50}.”

Comment 2: *The authors further suggest the generality of the approach, yet they explore a relatively limited scope of 2D perovskites that have not been assessed systematically by any outlined rationale. Although they summarize some of the materials considered (e.g., Figure 4a) later in the manuscript, there is no rationale behind the selection, which is relevant earlier in the discussion. Further clarification of the design principles for the systems is required, and, in particular, the rationale behind starting with the BA-GA system remains entirely unclear and needs to be elaborated on, as well as the focus on $n = 2$ compositions (e.g., Why this particular organic spacer and A cation? How does this help proceed with the objectives to clarify the structure-property relationships and the role of the inter/intra-layer interactions, particularly considering the relatively uncommon GA-based A-cation? What stimulates the $n = 2$ compositional focus, and would the approach apply to $n = 1$ compositions?).*

Response: We sincerely appreciate the reviewer for the comment. By carefully checking the literature, we have added a broad range of 2D perovskites with a large variety of aliphatic LA and A cations into our structural-property relationships, and further extended this principle from $n = 2$ to $n = 3$ and $n = 4$ 2D halide perovskites. The comparison between the previous version and the updated Figure 4a is depicted in Figure R2, where the newly added 2D perovskites are highlighted in red. The structural descriptor χ consistently demonstrates a strong correlation with the symmetric factor of PL peaks across an extensive set of a total 39 data points of 2D perovskites incorporating 6 different aliphatic LA spacers and 6 different A cations). Therefore, we are confident that the descriptor χ serves as an effective and universally applicable predictive parameter for the structural design of high-performance 2D perovskites.

Figure R2. (a) The previous version of Figure 4a, which shows the PL symmetric factor as a function of structural descriptor χ for various $n = 2$ RP perovskites under pressure regulation (BA-GA, HA-GA, and BA-MA) or with chemical tailoring [(LA)₂APb₂I₇, LA = CMA, HA, PA, BA, and iBA; A = GA, FA, and MA]. (b) The updated version of Figure 4a, where a large scope of 2D perovskites with various LA and A cations have been added into our structural-property relationships, and the principle has been successfully expanded to $n = 3$ and $n = 4$ 2D halide perovskites. The new added 2D perovskites are highlighted in red.

We focus on $n \geq 2$ 2D perovskites in this study because they possess additional intralayer structure (A cations), which are more structurally complex compared with $n = 1$ 2D perovskites. Extensive recent efforts have been made to tailor the interlayer LA spacer cations in $n = 1$ 2D perovskites^{10, 26-27}. While in the more structurally complex and tunable $n \geq 2$ 2D perovskites, no research to-date has considered the structural tuning and design of 2D perovskites from both inter- and intralayer perspectives to gain a more complete understanding for developing high-performance materials. In this work, we synergistically tuned both the inter- and intralayer structures (*i.e.* LA and A cations) of 2D perovskites, and engineered the excitonic properties towards desired optoelectronic performances. We introduced a new structural descriptor χ that considers the synergistic effects of both intra- and interlayer structural characteristics to elucidate the structure-property relationships in $n \geq 2$ 2D perovskites.

As for the choice of BA-GA $n = 2$ perovskite as the model compound for the high-pressure study, this is because the incorporation of the over-sized GA⁺ expands the perovskite cage, giving a notably large χ value to be 0.158 nm⁴ at ambient conditions (see Figure 4a). This leaves an ample room for pressure tuning of the structural descriptor χ , thereby enabling a

more wholistic investigation of structure-property relationship across a broader structural range.

The justification of the choice of BA-GA perovskite was added on Page 5 of the Revised Manuscript: “To uncover the intra and interlayer structural features that govern the excitonic properties, we start by examining the effect of pressure modulation on photoluminescence and photoconductivity of the 2D RP perovskite $(\text{BA})_2(\text{GA})\text{Pb}_2\text{I}_7$. We choose this perovskite because it incorporates an oversized A-cation (GA) into the perovskite cage, resulting in considerable octahedral distortion and a high level of exciton trapping and asymmetric PL peak. Additionally, the LA cations (BA) are flexible alkyl chains that make the interlayer is highly compressible. Therefore, $(\text{BA})_2(\text{GA})\text{Pb}_2\text{I}_7$ is an ideal perovskite to investigate whether pressure-induced structural changes can improve excitonic behavior and analyze the nuanced structural-property relationships across a vast parameter space.”

Comment 3: *In addition, the pressure levels used need to be put into context and related to the specific design for a better understanding (e.g., they start with a 2.1 GPa PL investigation without any rationale provided to follow the approach).*

Response: We would like to clarify that the experiment started from ambient condition, *i.e.* 0 GPa. The PL reaches its maximum at 2.1 GPa. We acknowledge that the Figure 2a could have potentially led to some confusion, which has been revised in the updated version shown below.

Updated Figure 2a. *In situ* steady-state PL spectra of BAGA at selected pressures starting from ambient pressure, where a remarkably enhanced emission with a more symmetric PL line shape is observed during compression, reaching the maximum at 2.1 GPa.

Comment 4: *In addition, the “synergistic effects” of intra- and inter-layer structures are emphasized to justify the approach, which is generally applicable beyond this particular study, as overall structural changes in layered perovskite materials result from the interrelated*

structural changes across layers of the hybrid perovskite structure. How is this distinctly considered within this study, and what justifies the emphasis on “synergistic effects” in this context?

Response: We thank the reviewer for the insightful comment. As discussed in the responses to Comment 2 above, distinctive from the $n = 1$ 2D perovskites, to enable a wholistic understanding of the optoelectronic properties for $n \geq 2$ 2D perovskites, the consideration of the “synergistic effects” of inter- and intralayer structures is necessary, because the intralayer A cations (absent from $n = 1$ 2D perovskites) influence the structures and properties.

To further illustrate the “synergistic effects” of intra- and interlayer structures, we step-by-step introduced a smaller MA and a shorter iBA to substitute the GA and BA cations in the intra and interlayer of $(\text{BA})_2(\text{GA})\text{Pb}_2\text{I}_7$, respectively, as shown in Figure R6a. The introduction of the smaller intralayer MA cation leads to a contraction of the perovskite cage in $(\text{BA})_2(\text{MA})\text{Pb}_2\text{I}_7$, while the substitution of the interlayer BA cation with the shorter iBA (more compact) results in the reduction of the interlayer distance in $(\text{iBA})_2(\text{GA})\text{Pb}_2\text{I}_7$, as depicted in Figure R6b. Consequently, the structural descriptor χ values of the two single-site-engineered compounds, $(\text{BA})_2(\text{MA})\text{Pb}_2\text{I}_7$ and $(\text{iBA})_2(\text{GA})\text{Pb}_2\text{I}_7$, reduce to 0.154 nm^4 and 0.151 nm^4 , respectively. Moreover, a further reduced structural descriptor χ can only be realized through the concurrent substitution of both intra and interlayer cations. Such synergistic tuning of intra and interlayer structures results in the final compound, $(\text{iBA})_2(\text{MA})\text{Pb}_2\text{I}_7$, which achieves a further reduced χ value to 0.149 nm^4 . It's noteworthy that among the four 2D perovskites examined, $(\text{iBA})_2(\text{MA})\text{Pb}_2\text{I}_7$ — exhibiting the lowest χ value — boasts the highest PL intensity and possesses a more symmetrical peak shape (Figure R6c). Therefore, such step-by-step cation engineering clearly demonstrates that enhanced optoelectronic performance can be achieved through the synergistic tuning of both intra and interlayer structures.

Figure R6. (a) Cation substitutions of GA to MA and BA to iBA that realize the synergistic tuning of intra- and interlayer structure. (b) Perovskite cage volumes and layer distances of various 2D perovskites with different LA and A cations, where the isoline of structural descriptor χ are shown as dash lines. (c) Steady-state PL spectra of the cation-substituted 2D perovskites, where the highest PL performance can be achieved in $(\text{iBA})_2(\text{MA})\text{Pb}_2\text{I}_7$, whose intra and interlayer structure is synergistically tuned to result in smaller structural descriptor χ .

Comment 5: In defining the structural descriptor, the authors also consider "non-hydrogen atoms", which should be carefully clarified in the main manuscript. In addition, they identify volume and interlayer distance as critical parameters for establishing the "correlation" with optoelectronic properties. How does this correlation apply to the simple lattice parameters (e.g., d -values) already representative of the overall structure? Why is pressure-induced change critical; are irreversible structural changes considered?

Response: Thanks for the comment. We focus solely on non-hydrogen atoms (*i.e.* C, N atoms) because hydrogen atoms take up much less room in the crystal structure and show little contribution to the electronic structure and dielectric constant of the interlayer ligands. As a result, their influence on this structure-property correlation in 2D perovskites is generally negligible.

To address the reviewer's next question, we have plotted the PL symmetric factors for various 2D perovskites as a function of the lattice parameter d -value of the structure. As shown in Figure R7, it is apparent that no clear correlation can be identified. The probable reason for the inadequate predicting power of simple lattice parameters is because they only depict the overall crystal structure and do not offer sufficient insights for the precise design of 2D perovskites. For example, HA and BA fall on different trendlines because HA has two extra C atoms and causes a larger d -value, but the behaviors of HA-GA and BA-GA are very similar.

Figure R7. A lack of correlation between PL symmetric factor as a function of a simple structure parameter d -value for various $n = 2$ RP perovskites under pressure regulation (BA-GA, HA-GA, and BA-MA) or with chemical tailoring [(LA)₂APb₂l₇, LA = CMA, HA, PA, BA, and iBA; A = ATA, GA, DMA, FA, and MA].

The intricate interplay between the intra- and interlayer structures requires a careful structural balance. In this work, the newly proposed structural descriptor χ considers the intra- and interlayer structural characteristics using perovskite cage volume (V) and packing

density (L/\sqrt{N}). The step-by-step correlation analysis for arriving at this structure descriptor χ ($V \times \frac{L}{\sqrt{N}}$) is shown in Supplementary Figure 14, where the χ possesses the strongest correlation with PL symmetric factors. Please refer to the detailed discussion from the response to Comment 4 of Reviewer 1. These updated correlation results clearly show that χ empirically correlates well with the symmetric factor of the PL peaks for 39 data points from a broad range of 2D perovskites under pressure tuning or with different compositions of LA and A cations (Figure 4a), and each component of χ (V , L , and \sqrt{N}) is essential for achieving such a correlation. Therefore, we believe that χ can serve as an effective predictive parameter for guiding the structural design towards high-performance 2D RP perovskites.

For the next question, we have conducted the PL spectra during compression and decompression, as shown in Figure R8. The PL energy can recover to its original value following the compression path, indicating the reversible structural changes upon compression.

Figure R8. (a) PL spectra and (b) corresponding PL peak positions of $(\text{BA})_2(\text{GA})\text{Pb}_2\text{I}_7$ during compression and decompression, where a completely reversible process can be found.

Comment 5: The literature is mostly comprehensive, yet the authors must put previous reports on pressure-induced transformations of layered hybrid perovskites into context. Similarly, they should refer to prior reports using PLQY as a predictive material design parameter.

Response: We appreciate the reviewer for the suggestion. Accordingly, we have cited more literatures that reports on the pressure-induced transformations of 2D halide perovskites in the Revised Manuscript^{4, 28-29} and further elaborated the context for the current work in the introduction on Page 3. We have also added the literatures that using PLQY as a predictive material design parameter in the Revised Manuscript^{26-27, 30-31}.

In summary, this is a very interesting and insightful analysis that provides important insights into the design of 2D perovskites. However, the approach requires more clarity and putting it

into context concerning prior work on the topic for a better understanding of the scope and impact. Finally, the authors should carefully address the design principles behind the model systems used for this purpose and the role of pressure in controlling the changes. I appreciate your consideration.

Response: We would like to express our gratitude to the reviewer for the valuable feedback and constructive comments on our manuscript. Each point has been meticulously addressed in our Revised Manuscript.

Reviewer #3

“Exciton engineering of 2D Ruddlesden-Popper perovskites by synergistically manipulating intra- and inter-layer structures” by S. Guo demonstrates that compressive strain can significantly modulate the optoelectronic properties in 2D perovskites. In particular, they observe a large increase in the brightness of PL, a change in shape (more symmetric) of the emission spectrum, and a red-shift as pressure is applied. This is coupled also with pressure-dependent changes in the absorbance profile, and photoconductivity of the 2D materials, which the authors ascribe to reductions in exciton binding energy pushing the material from mostly bound excitons, toward mostly free excitons, and eventually free charges. Based on these results, and on simulations which show how the compressibility of 2D perovskites is affected by the A-site cation and spacer cation, the authors choose a notoriously small A-site cation (FA, of FAPbI₃ perovskite fame) and a rigid spacer cation (CMA) to minimize a structural descriptor X they have developed. This new FA-CMA 2D R-P perovskite is structurally more similar to the strained perovskites they performed most of their characterization on, and as a result its emission (i.e., PLQY) is very bright. The experimental work and presentation are very clear and high-quality, though the concepts are not entirely novel as several other works have performed optical characterization of strained 2D perovskites: 10.1039/D2NR06816H, 10.1021/acseenergylett.7b00284, and the authors themselves have published adjacent papers in this space: 10.1126/sciadv.add1984. Still, I feel that the work is high quality, adds considerably to scientific understanding, and as such should be published.

Response: We deeply appreciate the reviewer’s positive comments and the recommendation to publish our manuscript in *Nature Communications*. We have included more discussion and citations to previous reports on (high pressure) studies of 2D perovskites to provide a more nuanced context and further distinguish the current study from previous reports in the introduction on Page 3, which is also in line with the requests by other reviewers.

Comment 1: *My one question I hope the authors consider in a minor revision is whether or not the mechanism behind the increased PL can be better understood—perhaps through*

calculation results that already exist. It strikes me that for PL to go up so strongly, simply freeing the exciton, or delocalizing it, can actually produce the opposite of the observed effect since this will allow excitons to diffuse further until they reach a trap site or annihilate with one another. Since PL is the result of radiative transitions winning a competition between themselves, non-radiative recombination, and dissociation, what I suspect is happening is that the oscillator strength of the exciton emission is being modulated significantly (in addition to its binding energy changing) and thus the radiation is becoming faster and competing better with non-radiative recombination which increases the brightness. This likely occurs because the compression (and indeed using FA-CMA) push the exciton hole and electron density closer and provide more spatial overlap for optical transitions to occur.

Response: We sincerely appreciate the reviewer for the insightful comment. We totally agree with the reviewer that excitons are more easily to be captured by traps when they become more delocalized. The emission enhancement can be ascribed to the accelerated radiative recombination, which more effectively rivals non-radiative recombination and dissociation processes. Such considerations are mostly consistent with the picture we present and enrich our discussion in this work.

To further clarify these, we have added the follow discussion about such perspectives on Page 8 of the Revised Manuscript: “The oscillator strength of the exciton emission undergoes significant modulation since it is proportional to the binding energy.³²⁻³³ Consequently, the radiative recombination of excitons becomes more efficient, enabling it to outperform non-radiative recombination, thereby leading to an increased PL intensity.”

References Cited in the Response Letter

1. Li, Q.; Zhang, L.; Chen, Z.; Quan, Z., Metal halide perovskites under compression. *J. Mater. Chem. A* **2019**, *7* (27), 16089-16108.
2. Zhang, L.; Wang, K.; Lin, Y.; Zou, B., Pressure Effects on the Electronic and Optical Properties in Low-Dimensional Metal Halide Perovskites. *J. Phys. Chem. Lett.* **2020**, *11* (12), 4693-4701.
3. Liu, S.; Sun, S.; Gan, C. K.; del Águila, A. G.; Fang, Y.; Xing, J.; Do, T. T. H.; White, T. J.; Li, H.; Huang, W.; Xiong, Q., Manipulating efficient light emission in two-dimensional perovskite crystals by pressure-induced anisotropic deformation. *Sci. Adv.* **2019**, *5* (7), eaav9445.
4. Yin, T.; Yan, H.; Abdelwahab, I.; Lekina, Y.; Lu, X.; Yang, W.; Sun, H.; Leng, K.; Cai, Y.; Shen, Z. X.; Loh, K. P., Pressure driven rotational isomerism in 2D hybrid perovskites. *Nat. Commun.* **2023**, *14* (1), 411.
5. Fu, R.; Zhao, W.; Wang, L.; Ma, Z.; Xiao, G.; Zou, B., Pressure-Induced Emission toward Harvesting Cold White Light from Warm White Light. *Angew. Chem., Int. Ed.* **2021**, *60* (18), 10082-10088.
6. Jaffe, A.; Lin, Y.; Mao, W. L.; Karunadasa, H. I., Pressure-induced conductivity and yellow-

to-black piezochromism in a layered Cu-Cl hybrid perovskite. *J. Am. Chem. Soc.* **2015**, *137* (4), 1673-8.

7. Guo, S.; Zhao, Y.; Bu, K.; Fu, Y.; Luo, H.; Chen, M.; Hautzinger, M. P.; Wang, Y.; Jin, S.; Yang, W.; Lu, X., Pressure-Suppressed Carrier Trapping Leads to Enhanced Emission in Two-Dimensional Perovskite (HA)₂(GA)Pb₂I₇. *Angew. Chem., Int. Ed.* **2020**, *59* (40), 17533-17539.

8. Li, X.; Fu, Y.; Pedesseau, L.; Guo, P.; Cuthriell, S.; Hadar, I.; Even, J.; Katan, C.; Stoumpos, C. C.; Schaller, R. D.; Harel, E.; Kanatzidis, M. G., Negative Pressure Engineering with Large Cage Cations in 2D Halide Perovskites Causes Lattice Softening. *J. Am. Chem. Soc.* **2020**, *142* (26), 11486-11496.

9. Katan, C.; Mercier, N.; Even, J., Quantum and Dielectric Confinement Effects in Lower-Dimensional Hybrid Perovskite Semiconductors. *Chem. Rev.* **2019**, *119* (5), 3140-3192.

10. Li, X.; Hoffman, J. M.; Kanatzidis, M. G., The 2D Halide Perovskite Rulebook: How the Spacer Influences Everything from the Structure to Optoelectronic Device Efficiency. *Chem. Rev.* **2021**, *121* (4), 2230-2291.

11. Zhang, L.; Wu, L.; Wang, K.; Zou, B., Pressure-Induced Broadband Emission of 2D Organic-Inorganic Hybrid Perovskite (C₆H₅C₂H₄NH₃)₂PbBr₄. *Adv. Sci.* **2019**, *6* (2), 1801628.

12. Fang, Y.; Zhang, L.; Yu, Y.; Yang, X.; Wang, K.; Zou, B., Manipulating Emission Enhancement and Piezochromism in Two-Dimensional Organic-Inorganic Halide Perovskite [(HO)(CH₂)₂NH₃]₂PbI₄ by High Pressure. *CCS Chem.* **2020**, *2*, 2203-2210.

13. Chen, Y.; Fu, R.; Wang, L.; Ma, Z.; Xiao, G.; Wang, K.; Zou, B., Emission enhancement and bandgap retention of a two-dimensional mixed cation lead halide perovskite under high pressure. *J. Mater. Chem. A* **2019**, *7* (11), 6357-6362.

14. Shi, Y.; Jin, Z.; Lv, P.; Wang, K.; Xiao, G.; Zou, B., Bandgap narrowing and piezochromism of doped two-dimensional hybrid perovskite nanocrystals under pressure. *J. Mater. Chem. C* **2023**, *11* (5), 1726-1732.

15. Geng, T.; Ma, Z.; Chen, Y.; Cao, Y.; Lv, P.; Li, N.; Xiao, G., Bandgap engineering in two-dimensional halide perovskite Cs₃Sb₂I₉ nanocrystals under pressure. *Nanoscale* **2020**, *12* (3), 1425-1431.

16. Liu, G.; Kong, L.; Guo, P.; Stoumpos, C. C.; Hu, Q.; Liu, Z.; Cai, Z.; Gosztola, D. J.; Mao, H.-k.; Kanatzidis, M. G.; Schaller, R. D., Two Regimes of Bandgap Red Shift and Partial Ambient Retention in Pressure-Treated Two-Dimensional Perovskites. *ACS Energy Lett.* **2017**, *2* (11), 2518-2524.

17. Jiang, Y.; Cui, M.; Li, S.; Sun, C.; Huang, Y.; Wei, J.; Zhang, L.; Lv, M.; Qin, C.; Liu, Y.; Yuan, M., Reducing the impact of Auger recombination in quasi-2D perovskite light-emitting diodes. *Nat. Commun.* **2021**, *12* (1), 336.

18. Xing, G.; Wu, B.; Wu, X.; Li, M.; Du, B.; Wei, Q.; Guo, J.; Yeow, E. K.; Sum, T. C.; Huang, W., Transcending the slow bimolecular recombination in lead-halide perovskites for electroluminescence. *Nat. Commun.* **2017**, *8*, 14558.

19. Anantharaman, S. B.; Jo, K.; Jariwala, D., Exciton-Photonics: From Fundamental Science to Applications. *ACS Nano* **2021**, *15* (8), 12628-12654.

20. Blancon, J. C.; Stier, A. V.; Tsai, H.; Nie, W.; Stoumpos, C. C.; Traore, B.; Pedesseau, L.; Kepenekian, M.; Katsutani, F.; Noe, G. T.; Kono, J.; Tretiak, S.; Crooker, S. A.; Katan, C.; Kanatzidis, M. G.; Crochet, J. J.; Even, J.; Mohite, A. D., Scaling law for excitons in 2D perovskite

quantum wells. *Nat. Commun.* **2018**, *9* (1), 2254.

21. Neutzner, S.; Thouin, F.; Cortecchia, D.; Petrozza, A.; Silva, C.; Srimath Kandada, A. R., Exciton-polaron spectral structures in two-dimensional hybrid lead-halide perovskites. *Phys. Rev. Mater.* **2018**, *2* (6).

22. Feldstein, D.; Perea-Causin, R.; Wang, S.; Dyksik, M.; Watanabe, K.; Taniguchi, T.; Plochocka, P.; Malic, E., Microscopic Picture of Electron-Phonon Interaction in Two-Dimensional Halide Perovskites. *J. Phys. Chem. Lett.* **2020**, *11* (23), 9975-9982.

23. Smith, M. D.; Karunadasa, H. I., White-Light Emission from Layered Halide Perovskites. *Acc. Chem. Res.* **2018**, *51* (3), 619-627.

24. Blancon, J. C.; Even, J.; Stoumpos, C. C.; Kanatzidis, M. G.; Mohite, A. D., Semiconductor physics of organic-inorganic 2D halide perovskites. *Nat. Nanotechnol.* **2020**, *15* (12), 969-985.

25. Fu, Y., Stabilization of Metastable Halide Perovskite Lattices in the 2D Limit. *Adv. Mater.* **2022**, e2108556.

26. Mauck, C. M.; Tisdale, W. A., Excitons in 2D Organic-Inorganic Halide Perovskites. *Trends Chem.* **2019**, *1* (4), 380-393.

27. Gong, X.; Voznyy, O.; Jain, A.; Liu, W.; Sabatini, R.; Piontkowski, Z.; Walters, G.; Bappi, G.; Nokhrin, S.; Bushuyev, O.; Yuan, M.; Comin, R.; McCamant, D.; Kelley, S. O.; Sargent, E. H., Electron-phonon interaction in efficient perovskite blue emitters. *Nat. Mater.* **2018**, *17* (6), 550-556.

28. Jaffe, A.; Mack, S. A.; Lin, Y.; Mao, W. L.; Neaton, J. B.; Karunadasa, H. I., High Compression-Induced Conductivity in a Layered Cu-Br Perovskite. *Angew. Chem., Int. Ed.* **2020**, *59* (10), 4017-4022.

29. Li, H.; Qin, Y.; Shan, B.; Shen, Y.; Ersan, F.; Soignard, E.; Ataca, C.; Tongay, S., Unusual Pressure-Driven Phase Transformation and Band Renormalization in 2D vdW Hybrid Lead Halide Perovskites. *Adv. Mater.* **2020**, *32* (12), e1907364.

30. Luo, H.; Guo, S.; Zhang, Y.; Bu, K.; Lin, H.; Wang, Y.; Yin, Y.; Zhang, D.; Jin, S.; Zhang, W.; Yang, W.; Ma, B.; Lu, X., Regulating Exciton-Phonon Coupling to Achieve a Near-Unity Photoluminescence Quantum Yield in One-Dimensional Hybrid Metal Halides. *Adv. Sci.* **2021**, *8* (14), e2100786.

31. Wang, Y.; Guo, S.; Luo, H.; Zhou, C.; Lin, H.; Ma, X.; Hu, Q.; Du, M. H.; Ma, B.; Yang, W.; Lu, X., Reaching 90% Photoluminescence Quantum Yield in One-Dimensional Metal Halide C₄N₂H₁₄PbBr₄ by Pressure-Suppressed Nonradiative Loss. *J. Am. Chem. Soc.* **2020**, *142* (37), 16001-16006.

32. Li, S.; Li, X.; Kocoj, C. A.; Ji, X.; Yuan, S.; Macropulos, E. C.; Stoumpos, C. C.; Xia, F.; Mao, L.; Kanatzidis, M. G.; Guo, P., Ultrafast Excitonic Response in Two-Dimensional Hybrid Perovskites Driven by Intense Midinfrared Pulses. *Phys. Rev. Lett.* **2022**, *129* (17), 177401.

33. Su, R.; Fieramosca, A.; Zhang, Q.; Nguyen, H. S.; Deleporte, E.; Chen, Z.; Sanvitto, D.; Liew, T. C. H.; Xiong, Q., Perovskite semiconductors for room-temperature exciton-polaritonics. *Nat. Mater.* **2021**, *20* (10), 1315-1324.

REVIEWER COMMENTS

Reviewer #1 (Remarks to the Author):

The work still lacks novelty, as basic concepts are already known (GA-BA composition: J. Am. Chem. Soc. 2020, 142, 26, 11486–11496; pressure-induced emission: <https://doi.org/10.1002/adv.202305597>,). Also, the descriptor is still pretty empirical and does not have much physical insight.

Reviewer #2 (Remarks to the Author):

The authors have addressed most of the reviewers' concerns, improving their report. However, several points are not adequately addressed and require detailed clarification to be able to recommend the manuscript for publication.

1) Firstly, the definition of “inter-“ and “intralayer” structures remains confusing and should be explicitly defined, particularly as it seems to reflect the structural elements associated with the spacer and the A-cation, while entirely neglecting the “interlayer” interactions within the organic spacer layer, arguing that “the effects of interactions between these cations are minimal” in response to the reviewer. This contradicts the bulk of the scientific literature on layered hybrid perovskites (and arguably their own observations) that suggests that tailoring the organic spacer has a profound effect on the resulting structure and optoelectronic characteristics despite most of these systems being linked to the inorganic framework via essentially the same (or comparable) ammonium groups. While Figure 1b helps put these notations into context more clearly, and it may be understandable that the authors neglect weaker interactions in their model to focus on the stronger ionic forces, this should be clarified in the manuscript and commented on from the perspective of potential limitations or scopes of the descriptor. For instance, the similar behaviour of HA-GA and BA-GA systems can be associated with their inter-spacer-layer conformation, which is, in turn, affected by the interactions between the organic spacers. It is misleading to suggest that these interactions do not play a role, particularly considering that the authors focus almost exclusively on alkylammonium spacers, which should also be justified in the introduction for clarity (Why not consider aromatic systems? This challenges the notion of the study being “comprehensive”, as suggested. Moreover, the cyclohexyl analogues should also be presented in the appropriate conformation to avoid confusion).

2) Similarly, the authors expand their scope of the systems analysed following the critical remarks, yet the comment on the clear rationale behind the selection of specific organic spacers is still lacking. In particular, they focus primarily on alkylammonium systems and entirely on the $n > 1$ compositions due to their “structural complexity and tunability” without putting the conventional 2D $n = 1$ systems into context and assessing the capacity of the descriptor to reflect their characteristics. This should be appropriately addressed in the discussion and defining the scope and rationale in the manuscript (not only responses to reviewers), even if the authors do not analyse $n = 1$ systems.

3) Finally, this reviewer still questions the use of the term “synergistic” in the context of this work. While some of the effects observed may be “synergistic” after all, the “synergistic tuning” or “synergistic manipulation” does not seem justified on the basis of simply using two different A/spacer cations, as in this regard, all 2D perovskites of $n > 1$ composition feature “synergistic effects”.

In general, this is very relevant work for the community, yet there is a lack of clarity on the scope of the effort that needs to be better put into context. While the authors attempted to address some of these points, such as by detailing their rationale in response to the reviewers, this was not reflected in clear amendments in the manuscript, leaving still some unintentional confusion that I hope these remarks help address.

I appreciate your consideration.

Point-by-point Responses to the Reviewers' Comments

(Manuscript ID: NCOMMS-23-34853B)

We would like to express our deep gratitude to the reviewers for dedicating their time and effort to review our manuscript. The insightful comments are very helpful for further improving the quality and clarity of our work. We provide a detailed responses to these comments below.

Reviewer #1

The work still lacks novelty, as basic concepts are already known (GA-BA composition: J. Am. Chem. Soc. 2020, 142, 26, 11486–11496; pressure-induced emission: <https://doi.org/10.1002/adv.202305597>). Also, the descriptor is still pretty empirical and does not have much physical insight.

Response: We respectfully disagree with Reviewer #1's comments. It appears that there may be a misinterpretation of the fundamental concepts of our work, as the references provided do not directly align with our key findings. The first reference focused on the synthesis of $(\text{BA})_2(\text{GA})\text{Pb}_2\text{I}_7$ without further structural tuning or property optimization, and the second reference concentrated on the $n = 1$ 2D perovskites where only the interlayer spacer (LA) cation was considered. It is crucial to note that our work delves into the more intricate and tunable $n > 1$ 2D perovskites, incorporating both intralayer and interlayer structural considerations (*i.e.* involving both LA and A cations). This comprehensive exploration of the intralayer and interlayer structures allows for a deeper understanding previously unavailable.

Regarding the reviewer's concern about the empirical nature of our structural descriptor, we would like to draw attention to the analogous situation in 3D perovskites where the widely utilized Goldschmidt tolerance factor is empirical but has demonstrated significant impact and wide acceptance by the research communities. Similarly, our newly proposed structural descriptor for 2D perovskites elucidates the relationship between structural characteristics and optoelectronic properties, offering an effective predictive parameter for guiding the structural design of novel 2D perovskites.

Reviewer #2

The authors have addressed most of the reviewers' concerns, improving their report. However, several points are not adequately addressed and require detailed clarification to be able to recommend the manuscript for publication.

Response: We appreciate the reviewer's recognition of our previous efforts and provide point-by-point responses to the remaining concerns below.

Comment 1: *Firstly, the definition of "inter-" and "intralayer" structures remains confusing and should be explicitly defined, particularly as it seems to reflect the structural elements associated with the spacer and the A-cation, while entirely neglecting the "interlayer" interactions within the organic spacer layer, arguing that "the effects of interactions between these cations are minimal" in response to the reviewer. This contradicts the bulk of the scientific literature on layered hybrid perovskites (and arguably their own observations) that suggests that tailoring the organic spacer has a profound effect on the resulting structure and optoelectronic characteristics despite most of these systems being linked to the inorganic framework via essentially the same (or comparable) ammonium groups. While Figure 1b helps put these notations into context more clearly, and it may be understandable that the authors neglect weaker interactions in their model to focus on the stronger ionic forces, this should be clarified in the manuscript and commented on from the perspective of potential limitations or scopes of the descriptor. For instance, the similar behavior of HA-GA and BA-GA systems can be associated with their inter-spacer-layer conformation, which is, in turn, affected by the interactions between the organic spacers. It is misleading to suggest that these interactions do not play a role, particularly considering that the authors focus almost exclusively on alkylammonium spacers, which should also be justified in the introduction for clarity (Why not consider aromatic systems? This challenges the notion of the study being "comprehensive", as suggested. Moreover, the cyclohexyl analogues should also be presented in the appropriate conformation to avoid confusion).*

Response: We would like to first clarify our statement in previous response letter that "the effects of interactions between these cations are minimal", which is in response to the reviewer's comment that "the authors exclude the organic **intralayer** interactions". Looking

back, we suspect that there may be a typo in the reviewer's use of “**intralayer**” and “**interlayer**” in the previous comments. Here, we would like to take this opportunity to clarify the contributions of both **intralayer** interaction and **interlayer** interaction.

The layers composed of $[\text{PbI}_6]^{4-}$ octahedra and A cations are designated as **intralayer**, and the LA cations situated between two neighboring $[\text{PbI}_6]^{4-}$ octahedral layers are referred as **interlayer**. In the intralayer, the A cations are separated by the Pb-I octahedra (*i.e.* enclosed in the perovskite cages), resulting in weak interactions between these A cations. That's why we stated that “*the effects of interactions between these cations are minimal*” in the previous response. On the other hand, we agree with the Reviewer that the interlayer interactions between the spacer cations could influence both structures and properties of 2D perovskites. These interactions include hydrogen bonding by halogen substitution¹⁻³, π - π interactions by introducing aromatic rings⁴⁻⁵, and the formation of intermolecular networks^{1, 6}. All these interactions are relatively strong and complicated, which will bring diverse effects on both structures and properties. Describing these diverse effects using a single structural parameter is a challenging endeavor, and thus far, the number of cases studied remains limited.

Therefore, in our study, we focused on the most extensively studied alkylammonium spacer cations without such strong interactions between the spacer cations, which comprise 40 cases to establish a quantitative structure-property relationship. Within these cases, our structural descriptor has already accounted for the contribution of the interlayer interactions between spacer cation to some degree, but such contribution is mostly reflected in the geometric consequences after the spacer cations pack into the interlayer space. Essentially, the descriptor characterizes the compactness of the spacer cation arrangement within the interlayer space (*i.e.* packing density, $\frac{L}{\sqrt{N}}$), also taking into consideration the intralayer factors for a comprehensive understanding.

To further clarify these points, we have added the definition of intralayer and interlayer of 2D halide perovskites on Page 4 of the revised Manuscript: “As shown in Figure 1b, the layers composed of $[\text{PbI}_6]^{4-}$ octahedra and A cations are termed as intralayer, whereas the LA cations situated between two neighboring $[\text{PbI}_6]^{4-}$ octahedral layers are referred as interlayer.”

The discussion on the interlayer interactions and the rationale behind the selection of specific organic spacers have been added on Page 16 of the Revised Manuscript: “We would like to point out that the interlayer interactions between organic spacer cations, including hydrogen bonding¹⁻³, π - π interactions⁴⁻⁵, and the formation of intermolecular networks^{1, 6}, could influence the structures and properties of 2D perovskites. All these interactions are relatively strong and complicated, which will bring diverse effects on both structures and properties. Describing these diverse effects using a single structural parameter is a challenging endeavor, and thus far, the number of cases studied remains limited. Therefore, here we focused on the extensively studied alkylammonium spacer cations without exhibiting such strong interactions to establish a quantitative structure-property relationship. In these cases, the structural descriptor χ has already included the contribution of the interlayer interactions between spacer cation to some degree, but such contribution is mostly reflected in the geometric consequences after the spacer cations pack into the interlayer space.”

Comment 2: *Similarly, the authors expand their scope of the systems analyzed following the critical remarks, yet the comment on the clear rationale behind the selection of specific organic spacers is still lacking. In particular, they focus primarily on alkylammonium systems and entirely on the $n > 1$ compositions due to their “structural complexity and tunability” without putting the conventional 2D $n = 1$ systems into context and assessing the capacity of the descriptor to reflect their characteristics. This should be appropriately addressed in the discussion and defining the scope and rationale in the manuscript (not only responses to reviewers), even if the authors do not analyze $n = 1$ systems.*

Response: Thanks for the suggestion. As for $n = 1$ 2D perovskites that lack perovskite cage, we have checked if part of the structural descriptor χ (*i.e.* packing density $\frac{L}{\sqrt{N}}$) can be applied to them. The PL symmetric factor against packing density for $n = 1$ systems is shown in Figure R1, where no clear correlation can be found. Note that various structural descriptors have been proposed for $n = 1$ perovskites, such as intra-octahedral distortion⁷, octahedral tilting⁸, and Pb displacement⁹, still waiting for a more general rule, but it is beyond the scope of our work.

For the more intricate and tunable $n > 1$ 2D perovskites, however, a suitable structural descriptor is still lacking but highly desired. In our work, we propose a new structural descriptor χ that considers both the intralayer and interlayer structural characteristics in the $n > 1$ 2D perovskites and establish a quantitative relationship between χ and PL properties. To define the scope of our study more clearly, we have added the following discussion on Page 13 of the Revised Manuscript: “Additionally, we tried to assess the applicability of this descriptor in $n = 1$ 2D perovskites, as shown in Supplementary Figure 16, where the absence of a clear correlation may be attributed to the omission of intralayer cations. Note that various structural descriptors have been proposed for $n = 1$ 2D perovskites⁸⁻¹⁰, which is beyond the scope of this work.”

Figure R1. The PL symmetric factor as a function of packing density $\frac{L}{\sqrt{N}}$ for various $n = 1$ 2D perovskites, where no clear correlation can be found.

Comment 3: Finally, this reviewer still questions the use of the term “synergistic” in the context of this work. While some of the effects observed may be “synergistic” after all, the “synergistic tuning” or “synergistic manipulation” does not seem justified on the basis of simply using two different A/spacer cations, as in this regard, all 2D perovskites of $n > 1$ composition feature “synergistic effects”.

Response: We appreciate the reviewer for bringing up this concern. The term “synergistic” refers to the interaction of two or more elements in a way that their combined effect is greater than the sum of their individual effects, which we believe can accurately capture the essence of our findings. In the case of step-by-step cation-substitution in the intralayer and interlayer

from $(\text{BA})_2(\text{GA})\text{Pb}_2\text{I}_7$ to BA-MA, iBA-GA, and further to iBA-MA (Figure R2a), the synergistic effect is demonstrated in both structure tuning and property optimization.

Figure R2. (a) Schematic illustration of intralayer and interlayer cation substitutions to achieve a synergistic structure and property tuning. (b) The values of structural descriptor χ for the various cation-substituted 2D perovskites, where concurrent intralayer and interlayer cation-substitution achieves the optimal χ value. (c-e) Relative PL intensity for various 2D perovskite systems with step-by-step cation substitutions. The PL intensity for concurrent intralayer and interlayer cation-substitution is larger than the sum of the PL intensities for individual intralayer or interlayer cation-substitution, further illustrating the "synergistic effect".

From the aspect of structure, when considering either intralayer or interlayer cation-substitution individually, there is limited tuning of the structural descriptor χ . Only the simultaneous substitution of both intralayer and interlayer cations achieves the optimized χ value (Figure R2b). From the aspect of optical properties, as shown in Figure R2c, isolated intralayer (BA-MA) or interlayer (iBA-GA) cation-substitution leads to PL enhancement of 14.3 or 2.6 times, respectively. However, a concurrent intralayer and interlayer cation-substitution (iBA-MA) results in a 27.9-times enhancement in PL, significantly surpass the sum of the individual effects (*i.e.*, $14.3 + 2.6 - 1 = 15.9$ times). Similar results have been demonstrated for other 2D perovskites, as shown in Figure R2 d&e. Thus, in both the structural tuning and

property optimization contexts, the collective effect of simultaneous intralayer and interlayer cation-substitution proves to be more significant than the sum of their individual contributions, illustrating the "synergistic effects". To further clarify the synergistic effects, we have added these results and discussion in the revised version of main text (on Page 15) and Supplementary Information (on Page 6 for the detailed discussion and Page 29 for Supplementary Figure 20).

In general, this is very relevant work for the community, yet there is a lack of clarity on the scope of the effort that needs to be better put into context. While the authors attempted to address some of these points, such as by detailing their rationale in response to the reviewers, this was not reflected in clear amendments in the manuscript, leaving still some unintentional confusion that I hope these remarks help address. I appreciate your consideration.

Response: We appreciate the reviewer's comments and the recognition of the contributions of our work to the community. In the Revised Manuscript, we have taken steps to enhance the clarity of the scope of our work. Specifically, we have incorporated the necessary clarifications and rationale into the main text of the Revised Manuscript, which was only shown in the "Point-by-point Responses to Comments" in the previous version. We sincerely appreciate the reviewer's feedback and the recognition of our responses.

References Cited in the Response Letter

1. Morteza Najarian, A.; Dinic, F.; Chen, H.; Sabatini, R.; Zheng, C.; Lough, A.; Maris, T.; Saidaminov, M. I.; García de Arquer, F. P.; Voznyy, O.; Hoogland, S.; Sargent, E. H., Homomeric chains of intermolecular bonds scaffold octahedral germanium perovskites. *Nature* **2023**, *620* (7973), 328-335.
2. Passarelli, J. V.; Mauck, C. M.; Winslow, S. W.; Perkinson, C. F.; Bard, J. C.; Sai, H.; Williams, K. W.; Narayanan, A.; Fairfield, D. J.; Hendricks, M. P.; Tisdale, W. A.; Stupp, S. I., Tunable exciton binding energy in 2D hybrid layered perovskites through donor-acceptor interactions within the organic layer. *Nat. Chem.* **2020**, *12* (8), 672-682.
3. Wu, Z.; Zhang, W.; Ye, H.; Yao, Y.; Liu, X.; Li, L.; Ji, C.; Luo, J., Bromine-Substitution-Induced High-Tc Two-Dimensional Bilayered Perovskite Photoferroelectric. *J. Am. Chem. Soc.* **2021**, *143* (20), 7593-7598.

4. Xue, J.; Wang, R.; Chen, X.; Yao, C.; Jin, X.; Wang, K.-L.; Huang, W.; Huang, T.; Zhao, Y.; Zhai, Y.; Meng, D.; Tan, S.; Liu, R.; Wang, Z.-K.; Zhu, C.; Zhu, K.; Beard, M. C.; Yan, Y.; Yang, Y., Reconfiguring the band-edge states of photovoltaic perovskites by conjugated organic cations. *Science* **2021**, *371* (6529), 636-640.
5. Gong, X.; Voznyy, O.; Jain, A.; Liu, W.; Sabatini, R.; Piontkowski, Z.; Walters, G.; Bappi, G.; Nokhrin, S.; Bushuyev, O.; Yuan, M.; Comin, R.; McCamant, D.; Kelley, S. O.; Sargent, E. H., Electron-phonon interaction in efficient perovskite blue emitters. *Nat. Mater.* **2018**, *17* (6), 550-556.
6. Ren, H.; Yu, S.; Chao, L.; Xia, Y.; Sun, Y.; Zuo, S.; Li, F.; Niu, T.; Yang, Y.; Ju, H.; Li, B.; Du, H.; Gao, X.; Zhang, J.; Wang, J.; Zhang, L.; Chen, Y.; Huang, W., Efficient and stable Ruddlesden–Popper perovskite solar cell with tailored interlayer molecular interaction. *Nat. Photonics* **2020**, *14* (3), 154-163.
7. Febriansyah, B.; Borzda, T.; Cortecchia, D.; Neutzner, S.; Folpini, G.; Koh, T. M.; Li, Y.; Mathews, N.; Petrozza, A.; England, J., Metal Coordination Sphere Deformation Induced Highly Stokes-Shifted, Ultra Broadband Emission in 2D Hybrid Lead-Bromide Perovskites and Investigation of Its Origin. *Angew. Chem., Int. Ed.* **2020**, *59* (27), 10791-10796.
8. Smith, M. D.; Jaffe, A.; Dohner, E. R.; Lindenberg, A. M.; Karunadasa, H. I., Structural origins of broadband emission from layered Pb-Br hybrid perovskites. *Chem. Sci.* **2017**, *8* (6), 4497-4504.
9. Han, X. B.; Jing, C. Q.; Zu, H. Y.; Zhang, W., Structural Descriptors to Correlate Pb Ion Displacement and Broadband Emission in 2D Halide Perovskites. *J. Am. Chem. Soc.* **2022**, *144* (40), 18595-18606.
10. Zhao, X.; Ball, M. L.; Kakekhani, A.; Liu, T.; Rappe, A. M.; Loo, Y. L., A charge transfer framework that describes supramolecular interactions governing structure and properties of 2D perovskites. *Nat. Commun.* **2022**, *13* (1), 3970.

REVIEWERS' COMMENTS

Reviewer #3 (Remarks to the Author):

After reviewing the full response letter to reviewers, and particularly the responses to reviewer 2, I found that the authors satisfactorily addressed the relevant concerns within the intended scope of the manuscript. There remain concerns about the applicability of the author's framework outside of non-aromatic RP organic ligands, but the manuscript now adequately sets the boundaries within which the framework applies and beyond which it may not. As such, I think the manuscript is suitable for publication.